Corrected: Publisher correction

# Spectroelectrochemical study of water oxidation on nickel and iron oxyhydroxide electrocatalysts

Laia Francàs[1,5]*, Sacha Corby[1,5], Shababa Selim[1], Dongho Lee[2], Camilo A. Mesa[1], Robert Godin [1,4], Ernest Pastor [1], Ifan E.L. Stephens [3], Kyoung-Shin Choi[2] & James R. Durrant [1]*

Ni/Fe oxyhydroxides are the best performing Earth-abundant electrocatalysts for water oxidation. However, the origin of their remarkable performance is not well understood. Herein, we employ spectroelectrochemical techniques to analyse the kinetics of water oxidation on a series of Ni/Fe oxyhydroxide films: FeOOH, FeOOHNiOOH, and Ni(Fe)OOH (5% Fe). The concentrations and reaction rates of the oxidised states accumulated during catalysis are determined. Ni(Fe)OOH is found to exhibit the fastest reaction kinetics but accumulates fewer states, resulting in a similar performance to FeOOHNiOOH. The later catalytic onset in FeOOH is attributed to an anodic shift in the accumulation of oxidised states. Rate law analyses reveal that the rate limiting step for each catalyst involves the accumulation of four oxidised states, Ni-centred for Ni(Fe)OOH but Fe-centred for FeOOH and FeOOH-NiOOH. We conclude by highlighting the importance of equilibria between these accumulated species and reactive intermediates in determining the activity of these materials.

[1] Department of Chemistry, Imperial College London, White City Campus, London W12 0BZ, UK. [2] Department of Chemistry, University of Wisconsin-Madison, Madison, WI 53706, USA. [3] Department of Materials, Imperial College London, South Kensington Campus, London SW7 2 AZ, UK. [4]Present address: The University of British Columbia, Department of Chemistry, 3247 University Way, Kelowna V1V 1V7 BC, Canada. [5]These authors contributed equally: Laia Francàs, Sacha Corby. *email: lfrancas@ic.ac.uk; j.durrant@imperial.ac.uk

The (photo)electrocatalytic oxidation of water is a key reaction for the development of more sustainable energy systems. It is pivotal in enabling the renewable synthesis of green hydrogen from water and/or the reduction of carbon dioxide to carbon-based fuels and chemicals. This oxidative process is one of the most kinetically and thermodynamically challenging reactions in sustainable fuels production. Ni and Fe oxyhydroxides are particularly promising catalysts in both alkaline electrolysers[1–4] and photoelectrocatalytic systems[5–10]. Such metal oxyhydroxides have layered structures with significant water permeability and varying degrees of long-range order, and exhibit complex electrochemical properties, including multiple, and relatively localised, redox states[11]. For such electrodes, it is therefore challenging to probe directly the densities and reaction kinetics of the oxidised states driving water oxidation under *operando* conditions using electrochemical data alone. Herein, we report a combined spectroelectrochemical analysis of water oxidation on a series of Ni/Fe oxyhydroxides. Employing these data, we determine the concentration of oxidised states driving water oxidation, and the rate constants and reaction orders for this reaction, thus providing new insights into both the reaction mechanism and the role of Fe and Ni centres in driving this catalysis.

NiOOH has been widely used and thoroughly studied as an electrode material in batteries[12–16]. Since the 1980s the use of NiOOH electrodes for water oxidation has received increasing interest[14]. In particular, Fe-incorporated NiOOH (Ni(Fe)OOH) is the most active earth abundant electrocatalyst for water oxidation under alkaline conditions[17,18]. It has been reported that even when pure NiOOH is targeted, Fe impurities in the electrolyte can spontaneously incorporate into NiOOH during the water oxidation reaction and considerably enhance catalytic performance[14,19]. The effect of this Fe incorporation can vary depending on the content and location of Fe in material, which in turn varies depending on the conditions of incorporation[20,21]. As a consequence, there is an intense debate on the identity of the species responsible for oxygen evolution in Ni(Fe)OOH, with no consensus thus far. Most DFT studies suggest that the oxygen–oxygen bond formation takes place at reactive Fe centres[20,22–25]. One study has reported the experimental detection of Fe(IV) species under catalytic conditions[26]. On the other hand, other studies could not detect such Fe species but rather observed Ni(IV) at high applied potentials[20,27,28]. Recent work has also directly related high-valence Ni centres with water oxidation catalysis[29]. In addition, Raman studies of Ni(Fe)OOH have detected active oxygen species related to the deprotonation of hydroxylated Ni surfaces[28]. The environment surrounding Fe atoms also has a structural impact, with a contraction of Fe–O bonds in Ni(Fe)OOH compared to FeOOH[20], which can be expected to affect the electronic and catalytic properties of the metal centre, as was recently reported by Hu et al. in a comparison of the activity of $Ni_xFe_{1-x}OOH$ with FeOOHNiOOH[30].

Recently, spectroelectrochemical techniques have been used to analyse Fe/Ni oxyhydroxides, reporting a 430 nm spectral feature when > 10% Fe was incorporated into NiOOH[31–33]. Herein, we employ *operando* voltage pulse-induced spectroelectrochemical and electrochemical current measurements to assay the concentrations of the oxidised species driving water oxidation catalysis and the resultant water oxidation reaction kinetics. We have previously demonstrated the potential of analogous photo-induced measurements on photocatalytic systems[34–39]. In this work, we apply such combined optical/electrochemical analyses to elucidate the reaction rate constants for an electrocatalytic system, focusing on high performance Fe/Ni oxyhydroxide electrocatalysts and in particular on the role of Fe/Ni metal centres in these materials. These analyses allows us to measure the

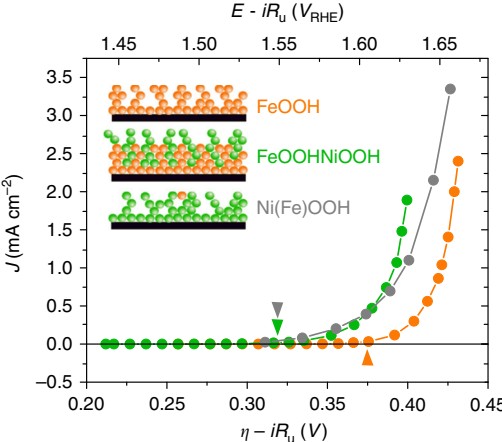

**Fig. 1** Steady-state *J–V* performance for each electrocatalyst. Steady state *J–V* curves at pH 13 (0.1 M NaOH aqueous solution) for FeOOH (orange), FeOOHNiOOH (green) and Ni(Fe)OOH (grey). These were recorded by measuring the steady-state current at different applied potential under atmospheric conditions. The FeOOH and Ni(Fe)OOH films are ~80 nm thick, while the FeOOHNiOOH film is ~120 nm thick (Supplementary Figs. 1 and 2). The overpotential is estimated as: $\eta = E_{RHE} - 1.23$ V and was corrected by the iR drop estimated by electrochemical impedance spectroscopy (EIS) measurements. Such *J–V* behaviour is typical of that reported previously for analogous Fe/Ni oxyhydroxide electrodes[19,28]. Arrows indicate the onset of the catalysis, which has been approximated to the potential at which the current reaches 20 μA cm$^{-2}$ Inset: Schematic representation of the different samples studied in this work. Orange represents an Fe-based group and green a Ni-based group

absorption spectra and concentrations of different oxidation states in three oxyhydroxide electrocatalysts with varying Fe/Ni content (Fig. 1, Inset), enabling direct measurement of catalytic turnover frequencies per accumulated oxidised state under catalytic conditions and thus providing new insights into their water oxidation catalysis.

## Results and discussion

**Sample preparation and electrochemical characterisation.** The three electrocatalysts studied in this work have been prepared following previously reported methods[5], and further characterisation is detailed in Table 1 (entries 1 and 2) and the "Methods" section plus Supplementary Figs. 1, 2 and Supplementary Table 1. The electrocatalysts can most simply be considered as pure FeOOH; FeOOHNiOOH in which FeOOH and NiOOH layers were deposited sequentially (atomic ratio of Fe:Ni = 0.79:0.21); and NiOOH with spontaneously incorporated iron—Ni(Fe)OOH (Fe:Ni 0.05:0.95)—as schematically depicted in Fig. 1 inset. For FeOOHNiOOH, SEM images (see Supplementary Fig. 2) indicate that NiOOH deposits inside the voids of the FeOOH layer, resulting in significant mixing of Fe and Ni at the FeOOH/NiOOH interface. These electrocatalysts have demonstrated high activity both as electrodes and, when deposited onto $BiVO_4$ and other semiconductors[5], as photoelectrodes for water oxidation. Herein we focus only on their electrocatalytic activity at pH 13, where they display the best activity.

Figure 1 presents a comparison of the steady-state current as a function of applied potential for the three electrocatalysts we studied (see also Table 1, entries 3 and 4. Tafel analyses and further information can be found in Supplementary Fig. 3 and are discussed in section "Mechanism discussion"). The anodic deposition method used in this study produces FeOOH and NiOOH without any further treatment. While FeOOH is stable,

**Table 1 Summary of data discussed in this work**

| Entry | | FeOOH | FeOOHNiOOH | Ni(Fe)OOH |
|---|---|---|---|---|
| | *Characterisation* | | | |
| 1 | Thickness (nm) | 80 | 120 | 80 |
| 2 | Fe (at.%) | 100 | 79 | 1 |
| 3 | Fe (at.%) in used samples | 100 | 79 | 5 |
| 4 | Catalysis onset[a] (overpotential, V) | 0.37 | 0.3 | 0.3 |
| 5 | Overpotential@2.5 mA cm$^{-2}$ (V) | 0.57 | 0.5 | 0.5 |
| 6 | $\varepsilon$ [+] (M$^{-1}$ cm$^{-1}$) | 2400 (500 nm) | 8250 (528 nm) | 8250 (528 nm) |
| 7 | $\varepsilon$ [++](M$^{-1}$ cm$^{-1}$) | 2400 (500 nm) | 3600 (550 nm) | 4000 (650 nm) |
| | *Kinetic data* | | | |
| 8 | $\tau$ (s)@1 mA cm$^{-2}$ | 0.76 (0.47 V) | 1 (0.44 V) | 0.46 (0.44 V) |
| 9 | TOF (s$^{-1}$)@1 mA cm$^{-2}$ | 0.3 | 0.25 | 0.5 |
| 10 | TOF (s$^{-1}$)@~340 mV overpotential | 0.003 | 0.021 | 0.070 |
| 11 | $\tau$ (s) from Figs. 3b and S15 | 0.8 (0.44 V) | 1 (0.42 V) | 0.6 (0.45 V) |
| 12 | $k_{obs}$ [cm$^6$ no. of e$^{-3}$ × s$^{-1}$] rate law plot | $1.2 \times 10^{-47}$ | $2.7 \times 10^{-48}$ | $5 \times 10^{-47}$ |
| 13 | % MOOH(++) of M-centres | 4 | 9 | 12 |

[a]Catalysis onset approximated to the potential at which the current reaches 20 µA cm$^{-2}$

NiOOH gradually converts to Ni(OH)$_2$ when exposed to air. This change can be easily recognised by the colour change from grey NiOOH to colourless Ni(OH)$_2$. Accordingly, electrocatalytic *J–V* data for Ni(Fe)OOH and FeOOHNiOOH were collected after activation to convert any Ni(OH)$_2$ to NiOOH (*J–V* curves before activation given in Supplementary Fig. 4). Activation consisted of five consecutive linear sweep voltammograms (LSV) from the open circuit potential (OCP) to 0.7 V of overpotential (see the "Methods" section for further details), which restored the grey coloured film (Supplementary Fig. 5), in agreement with previous reports[15,22,27,28]. As can be observed in Fig. 1, the activated Ni (Fe)OOH and FeOOHNiOOH electrodes have a very similar electrocatalytic performance, whilst FeOOH shows a 70 mV anodic shift in catalytic onset.

**Spectroelectrochemical characterisation.** Given the complex function of these electrocatalysts, we turned to *operando* optical absorption and electrochemical measurements to determine directly the concentration of oxidised species within each sample (see Supplementary Information for experimental details). For all three electrocatalysts, the optical absorption measurements were able to track two distinct oxidation processes. The first, pre-catalytic, oxidation process is, for simplicity, referred to herein as the oxidation of **MOOH(0)** to **MOOH(+)** (abbreviated to **(0/+)** where appropriate), and takes place during the activation process described above. The normalised differential absorption (ΔO.D.) spectra for this first oxidation are presented in Fig. 2a. Upon applying a more anodic potential, corresponding to conditions of catalytic water oxidation, a further oxidation process occurs, leading to the generation of doubly oxidised states, referred to herein as **MOOH(++)**. Note that this (+) and (++) notation is only intended to describe increased oxidation states and does not give any indication of the absolute oxidation state of the metal centres, which could only be obtained with a suite of complementary techniques beyond the scope of this study. The (+/++) oxidation causes a distinct optical absorption change for each electrocatalyst (Fig. 2b). It is apparent that the (+) and (++) ΔO.D. spectra are distinct for each sample (see Supplementary Fig. 6 for direct comparisons). For Ni(Fe)OOH, the first **(0/+)** oxidation has been assigned to the oxidation of Ni(OH)$_2$ to NiOOH, as observed by others[27], with the associated optical absorption increase assigned to nickel *d–d* interband transitions[28], although the details of this assignment are not critical for the study herein. For FeOOH, the **(0/+)** oxidation process is much more limited, oxidising only ca. 0.5% of the Fe atoms (Supplementary Table 2),

and we tentatively assign this process to the oxidation of defects, such as metal ions adjacent to oxygen vacancies. (Alternatively, this **(0/+)** process could involve the adsorption of water molecules onto coordinatively unsaturated surface Fe atoms, which several DFT-based studies attribute the first electron transfer to[20,40,41]). From the spectra in Fig. 2a, we can observe that the FeOOHNiOOH **(0/+)** oxidation results in a ΔO.D. spectrum that contains features from both FeOOH and Ni(Fe)OOH **(0/+)** oxidation. The NiOOH feature is more prominent, suggesting that despite comprising 79% Fe, the **(0/+)** oxidation process of FeOOHNiOOH is primarily associated with Ni(OH)$_2$ oxidation. In contrast, for the second **(+/++)** oxidation, observed under catalytic conditions, the ΔO.D. spectra are similar for FeOOH and FeOOHNiOOH, both exhibiting a peak around 400 nm (Fig. 2b). Small differences remain between them, indicating some impact of the presence of Ni on the **FeOOHNiOOH**(++) states, as we discuss further below. These spectra for FeOOH and FeOOHNiOOH are clearly different from that for the (+/++) oxidation in Ni(Fe)OOH, which presents a peak ca. 650 nm. This suggests that the **MOOH**(++) species accumulated in FeOOH and FeOOHNiOOH during water oxidation catalysis are similar in nature (and therefore both Fe-centred), whilst those accumulated in Ni(Fe)OOH are distinct, likely alluding to greater Ni character. To better determine whether this is the case, a NiOOH sample free of any Fe impurities was prepared and analysed in a purified electrolyte and otherwise comparable conditions (see the "Methods" section for details and Supplementary Figs. 8 and 9). As shown in Supplementary Fig. 8, the same two spectral features present for Ni(Fe)OOH are observed for the purified NiOOH sample: a **(0/+)** oxidation, assigned to the oxidation of any Ni (OH)$_2$; and a second **(+/++)** oxidation process, which gives rise to a peak at ca. 650 nm. As such, we can propose that **MOOH(++)** species accumulated under catalytic conditions in these mixed metal systems are Fe-centred at higher concentrations of iron, but Ni-centred at low Fe concentrations (≤5%). It may be the case that there are intermediate Ni/Fe ratios for which either both metal centres can accumulate charge. It is also possible that this charge becomes delocalised between centres, which would agree with spectra recently reported by Dau and coworkers for mixed Ni/Fe electrocatalysts, assigned to di-μ-oxo bridged Ni (IV)–Fe(III) motifs[32]. Hu and coworkers also propose dual-site water oxidation pathways with certain Ni/Fe oxyhydroxide structures[30], and a synergistic mechanism remains probable given the improved catalytic onset of FeOOHNiOOH compared to FeOOH, as discussed below.

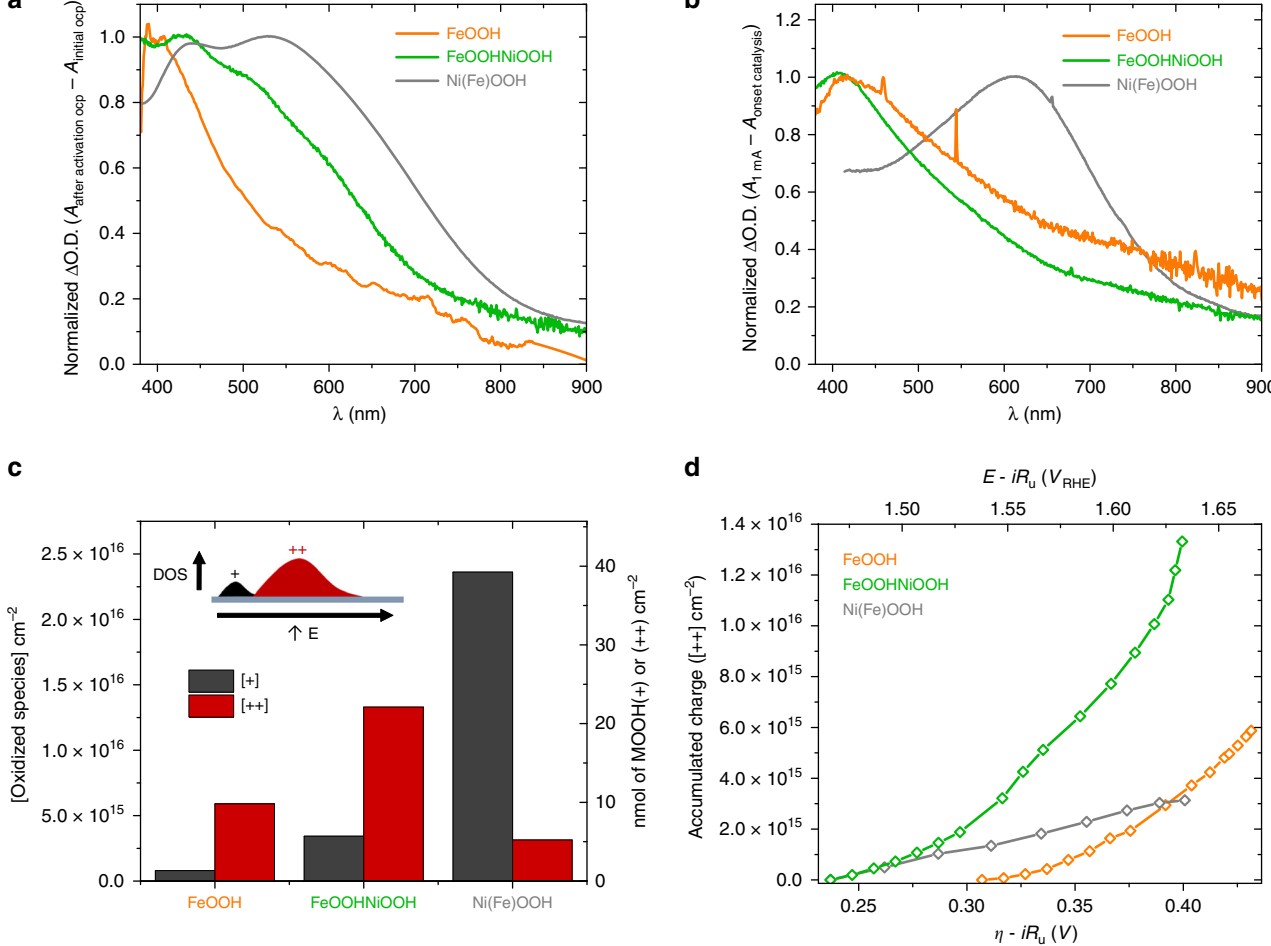

**Fig. 2** Accumulated (+) and (++) species for each electrocatalyst. **a** Normalised ΔO.D. spectra (to 1 at peak ΔO.D.) for the first (0/+), precatalytic, oxidation of FeOOH (orange), FeOOHNiOOH (green) and Ni(Fe)OOH (grey), obtained from the difference between the spectra at open circuit potential (OCP) after and before activation. **b** Normalised ΔO.D. spectra for the (+/++) oxidation wave correlated with water oxidation for FeOOH (orange), FeOOHNiOOH (green) and Ni(Fe)OOH (grey), obtained by subtracting spectra at OCP after activation from the spectra at 1 mA cm$^{-2}$ of current density. **c** Concentrations of oxidised **MOOH**(+) states generated in the activation process (black) and of **MOOH**(++) states generated at a current density of 1 mA cm$^{-2}$ (red) for each electrocatalyst. Inset: schematic representation of the corresponding densities of states (DOS) for these electrocatalysts. **d** Accumulated concentrations of **MOOH**(++) states vs. the applied overpotential (empty diamonds, solid lines). For both graphs FeOOH (orange), FeOOHNiOOH (green) and Ni(Fe)OOH (grey)

The extinction coefficients ($\mathcal{E}$) of all the involved species were estimated from ΔO.D., $J$–$V$ and step-potential spectroelectrochemistry (SP-SEC) data as detailed in Supplementary Information (Supplementary Figs. 10–12) and Table 1 (entries 6 and 7). Combining these extinction coefficients with our spectroelectrochemical data allows us to determine the concentrations of accumulated **MOOH**(+) and **MOOH**(++) species (Fig. 2c). We note that the magnitude of (0/+) oxidation is significantly higher in Ni(Fe)OOH compared to FeOOHNiOOH, although these two samples have a comparable amount of Ni (Supplementary Table 1). This trend is consistent with previous literature analyses which suggest that the presence of Fe stabilises Ni(OH)$_2$ relative to NiOOH[19,27]. Moreover, at current densities equal to 1 mA cm$^{-2}$, Ni(Fe)OOH exhibits the lowest concentration of catalytic **MOOH**(++) species and FeOOH-NiOOH the highest, as we discuss further below. The quantity of each **MOOH**(+) and **MOOH**(++) species is also provided as a percentage of the number of corresponding M metals (Supplementary Table 2).

Figure 2d plots the density of accumulated **MOOH**(++) species, measured as a function of applied potential for all three samples under conditions of electrocatalytic water oxidation. It is apparent that the Ni(Fe)OOH and FeOOHNiOOH electrocatalysts start accumulating **MOOH**(++) states ~70 mV negative of FeOOH, in agreement with their lower overpotential for current generation (compare Fig. 1 and Fig. 2d). When comparing the concentrations of accumulated **MOOH**(++) species (Fig. 2d), it is also apparent that FeOOHNiOOH accumulates the greatest number of **MOOH**(++) states, whilst both Ni(Fe)OOH and FeOOH accumulate far fewer **MOOH**(++) states. We observe that accumulated charge is not linear with applied potential for any of the catalysts, reflecting the non-uniform density of states common to these materials[22]. We note that **FeOOH**(++) states were not generated in non-aqueous electrolytes, highlighting the important role of water molecules and/or proton-coupled electron transfer in the generation of these states (see Supplementary Fig. 13). Upon addition of water, a very small oxidative wave can be determined, normally masked by the OER current in aqueous electrolytes, reaffirming the localised, molecular-style oxidations in these films. The concentrations of accumulated **MOOH**(++) species, ca. 10$^{16}$ cm$^{-2}$, are significantly greater than the density of metals on a flat MOOH surface, consistent with the

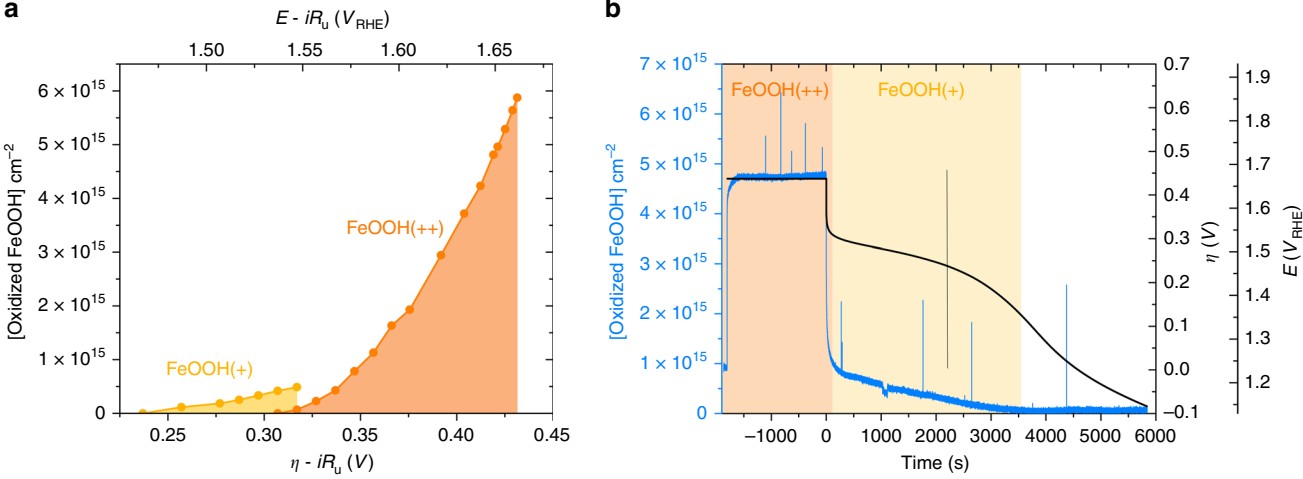

**Fig. 3** Accumulated **FeOOH(+)** and **FeOOH(++)** species with applied potential. **a** Accumulated **FeOOH(+)** (yellow) and **FeOOH(++)** (orange) concentrations vs. overpotential. **b** Simultaneous measurement of the 500 nm optical signal (blue trace), expressed as concentration, and electrode potential (black trace) induced by the application of a 0.44 V overpotential to FeOOH, and then switching to open circuit

previously reported permeability of such electrocatalysts to water, with the physical location of these accumulated species extending to some degree into the bulk of the material[42]. For example, accumulation of oxidised species beyond the atomic surface has been observed in the well-studied CoPi catalyst. In this case, bulk cobalt atoms were concluded to be catalytically active, leading to a proportionality between catalyst volume and activity[43,44]. The higher concentration of accumulated **MOOH(++)** states in FeOOHNiOOH, as well as the more negative onset potential, may be associated with extensive mixing of Ni and Fe at FeOOH/NiOOH interfaces (Supplementary Fig. 2). As such, in FeOOH-NiOOH it is possible that these states are associated with Fe atoms in proximity to Ni atoms of the NiOOH phase that stabilise **FeOOH(++)**[22,26].

Before analysing the water oxidation kinetics of these electrocatalysts, we considered further the potential dependence of **(0/+)** and the **(+/++)** oxidation processes, as illustrated in Fig. 3 using FeOOH as an example. As observed in Fig. 3a, the **(0/+)** oxidation begins to saturate at ~0.3 V, the same potential from which the **(+/++)** oxidation is initiated. For higher overpotentials, the concentration of **FeOOH(++)** states increases rapidly. Further insight into these two oxidation processes was obtained by measuring the electrode potential and optical absorption decays as a function of time, following a switch to open circuit conditions after applying a catalytic overpotential of 0.44 V (Fig. 3b). Switching to open circuit causes a sharp decay in the optical signal (<5 s), leaving a small residual signal. This rapid decay is assigned to **(++/+)** reduction associated with water oxidation, and the small residual signal to persistent **FeOOH(+)** states. Consistent with this assignment, the open circuit voltage also shows a rapid decay to +0.32 V, which corresponds to the onset of the **(+/++)** oxidation process. The residual optical and voltage signals then decay on the timescale of thousands of seconds, consistent with the slow relaxation of the activated **FeOOH(+)** catalysts back to the **FeOOH(0)** initial state. The concentration of residual **FeOOH(+)** states determined from the residual optical signal is $5 \times 10^{14}$ cm$^{-2}$, which is in excellent agreement with the magnitude of the **(0/+)** oxidation determined from the spectroelectrochemical data (Fig. 2c). Similar data and conclusions were obtained for the other two electrocatalysts, including UV–Vis–CV cycling of Ni(Fe)OOH (see Supplementary Figs. 14 and 15). Overall, these data confirm that the **(0/+)** process observed in our spectroelectrochemical data is an

oxidation process which, although important for catalytic function, is not directly involved in water oxidation catalysis. This is in accordance with other documentation in the literature for such mixed metal oxides reporting an activation or pre-catalytic step to bring the electrode to the catalytic resting state[27,45]. Consequently, we focus hereafter only on the analysis of the **(+/++)** oxidation observed under conditions of water oxidation catalysis.

**Kinetics of water oxidation.** By varying the potential applied, we can control the concentration of **MOOH(++)** states and measure the impact on the water oxidation kinetics. We note that an increased concentration of localised **MOOH(++)** states, correlated herein with accelerated water oxidation kinetics, also corresponds, from entropy considerations, to a higher free energy driving water oxidation. Similar to our previously reported kinetic analyses[34–39], we optically measured **MOOH(++)** state concentrations and electrochemically measured water oxidation current densities to determine how the lifetime, $\tau$ (s), of the **MOOH(++)** states driving water oxidation varies with **MOOH(++)** accumulation (Fig. 4a). We then used these data to generate a complementary plot of TOF (s$^{-1}$) of oxygen molecules per **MOOH(++)** (taking into account the need for four oxidised species to generate an oxygen molecule) against applied potential (Fig. 4b) (see Supplementary Methods for details of $\tau$ and TOF determination). Figure 4a shows the **MOOH(++)** lifetime, assigned to water oxidation kinetics, decreasing with increasing **MOOH(++)** concentration. The strong increase in **MOOH(++)** concentration for potentials up to 100 mV above the **(+/++)** accumulation onset potential (Fig. 2d), correlates with a rapid (circa two orders of magnitude) increase in TOF over this 100 mV potential range (Fig. 4b). We note that our studies determine the kinetics of the **MOOH(++)** species accumulated during steady-state water oxidation catalysis, but cannot resolve the kinetics of any shorter lived reactive intermediates in the catalytic cycle.

Figure 4a shows that at equal accumulated **MOOH(++)** concentrations, **Ni(Fe)OOH(++)** exhibits the fastest kinetics and **FeOOHNiOOH(++)** the slowest (by an order of magnitude). The slower water oxidation kinetics for **FeOOHNiOOH(++)** explain why this electrocatalyst exhibits a similar current density to Ni(Fe)OOH, despite a significantly higher concentration of

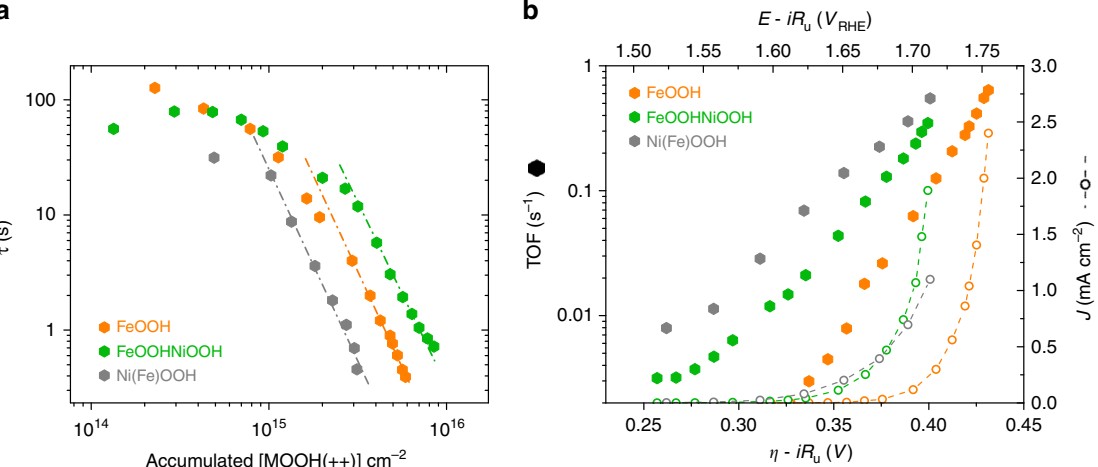

**Fig. 4** Kinetics of (++) species for each electrocatalyst. **a** Plot of the decay time ($\tau$) of accumulated **MOOH**(++) states against the accumulated **MOOH** (++) concentration (solid hexagons) for FeOOH (orange), Ni(Fe)OOH (grey) and FeOOHNiOOH (green). **b** Corresponding plots of TOF per O$_2$ generated against the applied overpotential (solid hexagons), overlaid with a plot of current density versus applied potential (empty circles, dashed lines)

**MOOH**(++) states. In other words, Ni(Fe)OOH has a lower concentration of accessible **MOOH**(++) states, but these states drive faster water oxidation catalysis, and the converse is true for FeOOHNiOOH. Figure 4b compares the TOFs as a function of the applied overpotential for the different electrocatalysts. Over the potential range examined, the TOF of Ni(Fe)OOH is faster than FeOOHNiOOH, and FeOOH is the slowest. The catalytic TOFs (s$^{-1}$) at 1 mA cm$^{-2}$ current density are indicated in Table 1, entry 9. From these data, we conclude that the more positive onset potential for water oxidation of FeOOH, in comparison to both Ni(Fe)OOH and FeOOHNiOOH, does not primarily result from slower water oxidation kinetics by **MOOH**(++) states, but rather from the anodic shift in the onset of accumulation of these states.

Similar time constants to those plotted in Fig. 4a were obtained directly from the decay time of the optical signal once the catalytic applied overpotential was turned off (see, for example, the fast optical decay phase in Fig. 3b; expanded data shown in Supplementary Figs. 16 and 18), as listed in Table 1, entries 11 and 12. This agreement in kinetics between our combined optical/ electrochemical data (Fig. 4a and Table 1, entry 8) and optical data alone (Supplementary Figs. 15 and 18 and Table 1, entries 11 and 12) strongly supports the validity of our analyses, as these two kinetic analyses employ very different experimental and data analysis procedures (see information in Supplementary Fig. 18 for more discussion). The TOFs determined from these data per **MOOH**(++) state are higher than those previously reported in the literature, which are typically determined per total mass or metal atoms density[19,21,27,46,47]. In contrast to those reports, herein we have quantified the concentration of the oxidised states driving water oxidation catalysis which, in general, is lower than the total content of electrochemically active metals. On the other hand, our TOF values are lower than those reported on Ni(Fe) OOH per Fe atom, suggesting that the Fe concentration in our samples is lower than the measured *operando* **MOOH**(++) state density, and as such, reaffirms that **Ni(Fe)OOH**(++) states are Ni-centred, as discussed above.

The FeOOHNiOOH film is likely to exhibit the most mixing of Fe and Ni and the largest FeOOH/NiOOH interfacial area. From these results reported herein, we find that this higher interfacial area does not improve the kinetics of the catalytic reaction, but does result in a cathodic shift of the onset of **MOOH**(++) accumulation when compared with pure FeOOH. This observation would agree with a higher degree of stabilisation of **FeOOH**

(++) when close to Ni atoms[22,26], thus facilitating charge accumulation but decreasing reactivity. The Ni(Fe)OOH sample shows enhanced kinetics and a similar onset of **MOOH**(++) accumulation as for FeOOHNiOOH, although Ni(Fe)OOH is not able to accumulate as large a concentration of **MOOH**(++) states. The faster water oxidation kinetics of Ni(Fe)OOH in conjunction with the accumulation of fewer **Ni(Fe)OOH**(++) species results in a similar overall performance to FeOOH-NiOOH. These differences in reactivity between FeOOHNiOOH and Ni(Fe)OOH suggest a different nature of the **MOOH**(++) species, in agreement with the different spectral features, attributed to Fe-centred vs. Ni-centred oxidised species, respectively. The overall result is then a comparable *J–V* curve for both samples.

**Mechanism discussion**. Analogous to the study of photoelectrodes such as haematite, we can analyse these catalysts under steady-state conditions using a rate law plot according to the following equation:[34]

$$\log J = \log k_{wo} + \alpha \log[MOOH(++)] \qquad (1)$$

where $J$ is the electrochemical steady-state current density (and therefore water oxidation flux), [MOOH(++)] is the concentration of **MOOH**(++) states, $k_{wo}$ is the water oxidation rate constant, and $\alpha$ the order of the reaction with respect to **MOOH**(++) concentration. From the gradient of the plots presented in Fig. 5a, we find that all three electrocatalysts exhibit a reaction order ($\alpha$) of approximately four under conditions of catalytic water oxidation (20 µA cm$^{-2}$–1 mA cm$^{-2}$). This order decreases at lower current densities (Supplementary Fig. 17), which we tentatively assign to an overlap with an alternative mechanistic regime of lower order (~order 1), as observed in several photoanode materials[36,37,39]. In other words, under operational conditions, the current density increases with the fourth power of the concentration of **MOOH**(++) states. These reaction orders were further confirmed by satisfactory fitting of the optical decay kinetics presented in Fig. 3b and Supplementary Fig. 15 with a model incorporating first and fourth-order reactions (Supplementary Fig. 18). This observation indicates that four **MOOH**(++) species are required to overcome the rate-determining step of the catalysis (RDS). Interestingly, the observation of the same order of reaction for all electrocatalysts, and the resulting conclusion of an inherently similar reaction mechanism between them, contrasts with our Tafel analyses of

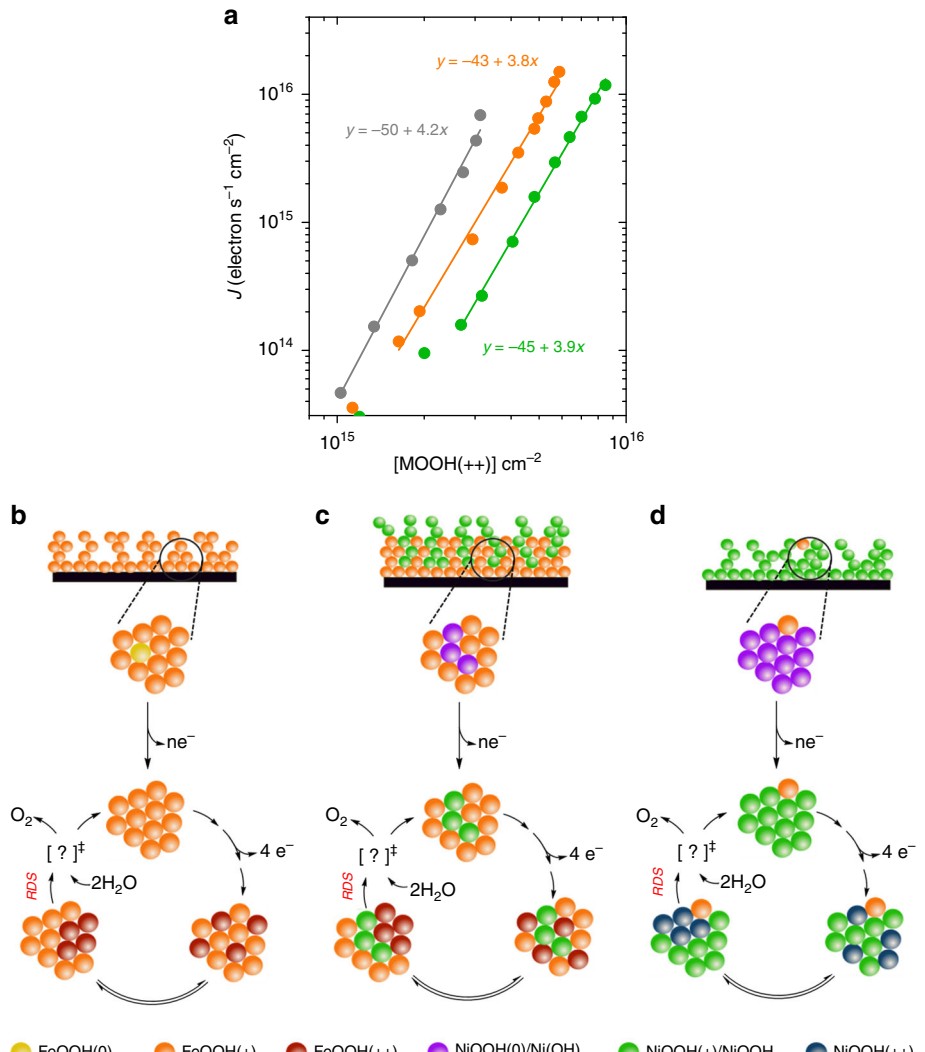

**Fig. 5** Mechanistic inights on the WO mechanism. **a** Rate law plots of the water oxidation current density versus the concentration of **MOOH(++)** states for FeOOH (orange), Ni(Fe)OOH and FeOOHNiOOH (green). **b–d** Simple schematic representations of the water oxidation process based on the experimental observations herein for (**b**) FeOOH, (**c**) Ni(Fe)OOH and (**d**) FeOOHNiOOH. Further details of shorter-lived intermediates are beyond the focus of this study. Colour code: **FeOOH(0)** corresponding to reduced defect states in FeOOH (yellow spheres); **FeOOH(+)** (orange spheres); **FeOOH(++)** (red spheres); **NiOOH(0)** corresponding to $Ni(OH)_2$ (purple spheres); **NiOOH(+)** corresponding to NiOOH (green spheres); **NiOOH(++)** (blue spheres)

the same electrodes, which yield significantly differing slopes (Supplementary Fig. 3), often interpreted as indicating different reaction mechanisms[48]. This disparity can be attributed to charge accumulation in these materials corresponding to multiple, localised redox oxidations at sites of direct electrolyte contact and therefore not behaving as ideal metals. As such, the different Tafel slopes most likely originate from differences in the dependence of charge density upon applied potential (Fig. 2d), rather than being indicative of different reaction mechanisms. The rate law data also confirm the kinetic trends discussed above, with the rate constant for water oxidation catalysis decreasing in the order **Ni(Fe)OOH(++)** > **FeOOH(++)** > **FeOOHNiOOH(++)** (see Table 1, entry 12). We note the order of reaction observed for these electrocatalysts was found to be pH sensitive, changing to three when the electrocatalysts are studied at pH 7 (see Supplementary Fig. 19). This is consistent with key role of protons in water oxidation as observed for photosynthetic water oxidation, although a full analysis of the effect of pH upon this electrocatalysis is beyond the scope of this study.

Figure 5b–d presents a simple mechanistic scheme based on our results. For all three electrocatalysts, the **(0/+)** process takes place before catalysis, corresponding to the oxidation of defect sites (most likely oxygen vacancy states) for FeOOH or the oxidation of $Ni(OH)_2$ to NiOOH for FeOOHNiOOH and Ni(Fe)OOH. After this first **(0/+)** oxidation process, the second oxidation **(+/++)** takes place, which correlates with the onset of electrocatalytic water oxidation. As illustrated in Fig. 5b–d, the **MOOH(++)** states generated can be considered in equilibrium with much shorter lived reactive intermediates (which are not directly observed in this study), with four such species being required to overcome the RDS. This fourth-order behaviour is consistent with published DFT work suggesting that the RDS corresponds to the extraction of the fourth electron by an already oxidised neighbouring species, from a bimetallic cluster, indicative of a cooperative mechanism (as illustrated in Supplementary Fig. 21)[40]. This high order of reaction would also agree with reports suggesting that the RDS is related with oxygen release[27], and the reported observation of accumulated oxidised species in XAS experiments prior to O–O bond formation[49].

The nature of the states driving water oxidation catalysis in Ni(Fe)OOH, and the role of Fe incorporation into NiOOH, is one of the largest controversies in this field. Previous studies have suggested that either Fe[18,20,22,26,50] or Ni[28,29,51] centres form the catalytic sites for water oxidation in such electrocatalysts. We note that our study focuses only on monitoring the oxidised states which accumulate to drive water oxidation catalysis, rather than directly probing the reactive intermediates involved in the oxygen–oxygen bond formation, and the rate constant we determine is with respect to these accumulated states. As discussed above, the different ΔO.D. spectra for **Ni(Fe)OOH(++)** and **FeOOHNiOOH(++)** states suggest these oxidised states accumulated under catalytic conditions are different in nature. These spectra suggest that **Ni(Fe)OOH(++)** states are associated either with oxidation of sparsely incorporated Fe within a NiOOH environment, or with Ni groups. Given the smaller quantity of accumulated (++) species in this sample (Fig. 2c), one could assume that these species are Fe-centred consistent with the smaller concentration of iron present. However, the concentration of **Ni(Fe)OOH(++)** states is at least twice the total concentration of Fe in the sample (see Supplementary Information and Table 1, entries 3 and 13), indicating a greater probability that these **Ni(Fe)OOH(++)** states are centred on Ni atoms. Furthermore, additional analysis of an NiOOH electrocatalyst kept free of any Fe impurities resulted in similar spectral features, thus effectively ruling-out any involvement of Fe centres in **MOOH(++)** species accumulation in samples with low concentrations of iron. This is in agreement with the detection of $Ni^{4+}$ (i.e. **NiOOH(++)** in our nomenclature) in XAS and XANES experiments on this type of catalyst[27,28]. $^{18}O$-labelling experiments by Hu and coworkers also indicate different active sites for Ni and NiFe oxides (Fe ≈ 25%)[52], as do light scattering experiments on similar samples by Dau and coworkers, in-line with our conclusions herein[53]. However, the Fe concentration at which the favoured **MOOH(++)** species accumulated switches from Ni-centred to Fe-centred remains to be determined. It also cannot be ascertained from these data alone whether a regime exists in which both metals are responsible for charge accumulation, as reported by others[30,32,54]. A degree of synergism between metal centres is probable, considering the improved charge accumulation found in the FeOOHNiOOH sample over the FeOOH sample. More experiments are being carried out in our group in order to shed light on this question.

Our experiments demonstrate that the atomic nature of the states accumulated during the steady state of water oxidation catalysis on Ni–Fe oxyhydroxides can change depending on the amount and method of Fe incorporation into NiOOH. We find that FeOOHNiOOH, which exhibits large islands of FeOOH phase, favours a catalytic process driven by Fe-centred oxidised states, most probably due to the easier accumulation of **FeOOH(++)** species at the interface between NiOOH and FeOOH phases. On the other hand, when the Fe is spontaneously incorporated in low amounts into NiOOH, charge accumulation is favoured on Ni centres, resulting in a lower accumulation of the oxidising equivalents driving water oxidation, balanced by faster water oxidation kinetics. These results could thus help to reconcile the seemingly discordant experimental results found in the literature, highlighting the need of comparable experimental procedures in order to study these types of electrocatalysts.

## Methods

**Electrocatalyst synthesis**. The three electrocatalysts studied in this work (FeOOH, Ni(Fe)OOH, and FeOOHNiOOH) were anodically electrodeposited on fluorine-doped tin oxide (FTO) (Hartford Glass) substrates. An undivided three-electrode cell consisting of an FTO working electrode (WE), a Ag/AgCl (4 M KCl) reference electrode (RE), and a platinum counter electrode (CE) were used for

electrodeposition. The platinum CE was prepared by sputter coating 100 nm of platinum on top of a 20 nm titanium adhesion layer on a clean glass slide. A VMP2 multichannel potentiostat (Princeton Applied Research) was used to perform the electrodeposition. All aqueous solutions used in this study were prepared using DI water further purified by a 3-module water purification system (Barnstead E-pure water purification systems). The resistivity of the output water was ≥ 18 MΩ cm.

The anodic deposition of FeOOH was carried out in a 10 mM $FeSO_4 \cdot 7H_2O$ (Sigma-Aldrich, 99%) aqueous solution (40 mL) (pH ~ 4.8 as prepared) at 70 ºC after purging the solution with $N_2$ for 30 min. The deposition was performed at 1.2 V versus the Ag/AgCl RE with gentle stirring immediately after $FeSO_4 \cdot 7H_2O$ was fully dissolved. The total charge passed was 23 mC/cm². This resulted in an ~80 nm-thick FeOOH film (Supplementary Fig. 1a, b). The lateral dimensions of the FeOOH film (and all other electrocatalyst films prepared in this study) were 1 cm × 1.2 cm. The as-prepared light-orange FeOOH film was rinsed with DI water and dried with a gentle stream of air. Then, the film was air-dried overnight before either NiOOH electrodeposition or spectroelectrochemical measurements were performed.

NiOOH was anodically deposited from a 100 mM $NiSO_4 \cdot 6H_2O$ (Sigma Aldrich, 98%) aqueous solution (40 mL) at 70 ºC. The pH was adjusted to 6.8−7.0 using a 0.070 M NaOH (Sigma Aldrich, 97%) aqueous solution prior to raising the temperature. The deposition was carried out at 1.2 V versus the Ag/AgCl RE with gentle stirring. The as-prepared NiOOH film was rinsed with DI water and dried with a gentle stream of air. Then, the film was air-dried overnight before electrochemical measurements were performed. The as-deposited NiOOH was black but it gradually became transparent due to the conversion of NiOOH to Ni(OH)₂. The total charge passed to deposit a NiOOH film with a thickness comparable to that of the FeOOH film was 150 mC/cm². When the FeOOH and NiOOH films were prepared with comparable thicknesses, inductively coupled plasma optical emission spectrometry (ICP–OES) (Perkin Elmer Optima 2000) results showed that the amount of Ni present in the NiOOH film was ~1/5 the amount of Fe present in the FeOOH film (Supplementary Table 1). This is because NiOOH is composed of vertically oriented, loosely packed, larger sheets while FeOOH is composed of more densely packed smaller plates (Supplementary Fig. 1). Since the plating solution used for the NiOOH deposition contained Fe impurities from the Ni source ($NiSO_4 \cdot 6H_2O$, 98%), the as-prepared NiOOH contained 1 at.% Fe (Supplementary Table 1). Also, since the electrolyte used to investigate the oxygen evolution properties of NiOOH (0.1 M NaOH) contained Fe impurities from NaOH (Aldrich, 98%), additional Fe was incorporated into NiOOH (5 at.% Fe) during the investigation (Supplementary Table 1). Therefore, NiOOH used in this study is denoted as Ni(Fe)OOH.

To investigate NiOOH without any Fe impurities, samples were prepared in the same manner as Ni(Fe)OOH but with two modifications. First, a higher purity $NiSO_4 \cdot 6H_2O$ (Sigma Aldrich, 99.998%) was used to make a plating solution. (Low purity Ni sources can contain Fe as an impurity.) Second, the pH of the plating solution was not further adjusted because NaOH added to adjust the pH can also contain Fe impurities. The pH of the as-prepared 100 mM $NiSO_4 \cdot 6H_2O$ aqueous solution was 6.10. The absence of Fe in the NiOOH film prepared from this plating solution was confirmed by ICP–OES. All subsequent electrochemical analyses of the Fe-free NiOOH samples were performed in purified electrolyte, in accordance with a procedure reported by Trotochaud et al.[19], to maintain the pure NiOOH.

Finally, the FeOOHNiOOH film was prepared by sequential deposition of FeOOH and NiOOH layers using the methods described above (23 mC/cm² was passed to deposit FeOOH and 150 mC/cm² was passed to deposit NiOOH). Although the same charge was passed to deposit NiOOH in both Ni(Fe)OOH and FeOOHNiOOH samples, the amount of Ni present in FeOOHNiOOH was slightly greater than that in Ni(Fe)OOH because the deposition of NiOOH on FeOOH is easier than that on FTO (Supplementary Table 1). The resulting FeOOHNiOOH film was ~120 nm thick (Supplementary Fig. 2), which is less than the sum of the FeOOH and NiOOH layers prepared separately. Also, when NiOOH was deposited on FeOOH, the change in surface morphology and film thickness was not significant until 2/3 (100 mC/cm²) of the total charge was passed (Supplementary Fig. 2). This suggests that a considerable amount of NiOOH was deposited to fill the voids in the FeOOH layer and significant mixing of Ni and Fe was achieved in FeOOHNiOOH. The as-prepared FeOOHNiOOH was rinsed with DI water and dried with a gentle stream of air. Then, the film was air-dried overnight before electrochemical measurements were performed. The atomic ratio of Fe:Ni in FeOOHNiOOH was 79:21 (Supplementary Table 1). The Fe content in FeOOHNiOOH also increased during the electrochemical investigation of this film in 0.1 M NaOH. However, the relative amount of Fe incorporated into FeOOHNiOOH was negligible compared with the original amount of Fe present. Therefore, the atomic ratio of Fe:Ni was unaffected during electrochemical investigation.

**Electrochemical set-up**. Electrochemical experiments were carried out using an Autolab potentiostat (PGSTAT 101) and a typical three-electrode configuration with a platinum mesh as counter electrode, an Ag/AgCl electrode (saturated KCl) as reference and the electrocatalysts as working electrode. A 0.1 M sodium hydroxide (pH 13) was used as the electrolyte in all experiments, other than the analysis in Supplementary Fig. 19, which used 0.1 M potassium phosphate buffer

(pH 7). The measured values (vs. Ag/AgCl) were then converted to potentials against the reversible hydrogen electrode (RHE) using the Nernst equation:

$$V_{RHE} = V_{Ag/AgCl} + V^0_{Ag/AgCl} + 0.059 \cdot pH \qquad (S2)$$

$$V^0_{Ag/AgCl}(\text{saturated KCl}) = 0.199 \, V$$

All the potentials reported in this work have been corrected for the uncompensated series resistance (33–44 Ω) after data collection. This resistance was estimated from electrochemical impedance (EIS) measurements at high applied potentials, as the series resistance of the system. This series resistance was estimated by fitting the data collected from 0.1 MHz to 1 Hz and fitted using the Randles circuit model.

**Spectroelectrochemical experiments**. SEC, i.e. optical absorption as a function of applied potential, was measured by fitting the spectroelectrochemical cell in a Cary 60 UV–Vis spectrometer (Agilent Technologies). The measured data is generally presented as spectroelectrochemical difference spectra (ΔO.D.), which are generated by subtracting a reference spectrum (usually at the OCP or water-oxidation onset) from the absorption data obtained under conditions of interest (e.g. at current densities of 1 mA cm$^{-2}$, as in Fig. 2b). The technique is explained in detail by Pastor et al. [35].

**Step-potential spectroelectrochemistry**. We used this technique to estimate the extinction coefficient of the doubly oxidised species. This technique uses an electrochemical pump and an optical probe. The electrochemical pump (step-potential) is carried out by applying a squared (ON/OFF) voltage (potential difference) until steady-state conditions are reached. The effect of the applied potential on the electrocatalyst was monitored using light from a 100 W tungsten lamp (Bentham IL1), equipped with an Oriel cornerstone 130 monochromator. The transmitted probe light was filtered by several band pass and long pass filters (Comar Optics) and detected by a silicon photodiode (Hamamatsu S3071). Collected photons were converted to a voltage signal, sent to an amplifier (Costronics) and recorded by an oscilloscope (Tektronics TDS 2012c) and with a DAQ card (National Instruments, NI USB-6211) on the timescale of ms–s. Simultaneously, the extracted current was monitored using a Palmsens3 potentiostat. All data were acquired on home-built LabView software.

**Reporting summary**. Further information on research design is available in the Nature Research Reporting Summary linked to this article.

## Data availability
The data from which Figs. 4 and 5a are constructed are presented in the Supplementary Information. All other raw data can be made available upon reasonable request.

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

## Acknowledgements

J.R.D. acknowledges financial support from the European Research Council (project Intersolar 291482) and EPSRC project M_RHEX (EP/R023581/1). This project has received funding from the European Union's Horizon 2020 research and innovation programme under grant agreement 732840-A-LEAF. C.A.M. thanks COLCIENCIAS (call 568) for funding, L.F. thanks the EU for a Marie Curie fellowship (658270), E.P. and S.S. thanks the EPRSC for a DTP scholarship and S.C. thanks Imperial College London for a Schrödinger Scholarship. K.-S.C. acknowledges financial support by the National Science Foundation (NSF) under the NSF Center CHE-1764399.

## Author contributions

Conceptualisation: L.F., K.-S.C., J.R.D.; Sample synthesis: D.L.; Data collection/Analysis: L.F., S.C., S.S.; Data fittings: C.A.M., R.G.; Experimental design/Discussion; L.F., S.C., S.S., C.A.M., E.P, I.E.L.S, J.R.D.; Visualisation: L.F., S.C.; Writing: L.F., S.C.; Funding acquisition: K.-S.C., J.R.D.

## Competing interests

The authors declare no competing interests.
