## [Peer Review File · Nature Communications]

Reviewers' comments:

Reviewer #1 (Remarks to the Author):

Summary:

The manuscript deals with UV-vis spectroscopic investigations of electrodeposited Ni-Fe(OxHy) catalysts, currently the most promising catalyst for the alkaline water oxidation reaction (OER). The nature of the metal redox states during applied catalytic potential is still under debate owing to a complex redox cycle that extends into the OER regime. This manuscript provides new insight into this complexity of the redox system by utilizing time-resolved in situ UV-vis spectroscopy. The experiments are well performed and the results reveal new surprising insights, where two spectroscopic fingerprints are discerned; both Ni-centred oxidation (the [+] state) prior to OER and Fe-centred metal oxidation hidden in the catalytic regime (the [++] state). Further kinetics are resolved by evaluating kinetic time traces, and information is obtained regarding water oxidation kinetics. However, there are some question marks that need to be thoroughly addressed, and a few already published work on time-resolved UV-vis spectroscopy (and time-resolved X-ray absorption spectroscopy) on Ni-Fe(OxHy) catalysts has not been recognized in the current manuscript.

I would recommend this manuscript for publication since it provides a new detailed and unique set of data along with a clarification of metal-centred redox states in Ni-Fe catalysts. However, I would recommend the authors to carefully address the specific concerns before a final decision is reached.

Comment #1

Make sure that relevant literature on the topic is recognized and cited in the manuscript. The articles listed below deal with "dynamic" and time-resolved in situ UV-vis or X-ray absorption spectroscopic measurements of Ni-Fe(OxHy) catalysts. Rates for the metal oxidation/reduction process are discussed in the in situ UV-vis study by Loos et al. [1], both during CV-cycling and similar potential jumps as presented in your study. The rate constants of the oxidation processes from in situ XAS data are presented in the work of González-Flores et al.[4] which includes both CV cycling and potential jumps. In the work of Görlin et al. [2-3] in situ UV-vis data is presented during CV-cycling of both Ni-Fe catalysts and of physically mixed Ni+Fe catalysts.

Work on in situ UV-vis studies of Ni-Fe(OxHy) catalysts:

1) Loos, S., Zaharieva, I., Chernev, P., Lißner, A. & Dau, H. Electromodified NiFe Alloys as Electrocatalysts for Water Oxidation: Mechanistic Implications of Time-Resolved UV/Vis Tracking of Oxidation State Changes. *ChemSusChem* 0, (2019).

2) Görlin, M. et al. Tracking Catalyst Redox States and Reaction Dynamics in Ni-Fe Oxyhydroxide Oxygen Evolution Reaction Electrocatalysts: The Role of Catalyst Support and Electrolyte pH. *J. Am. Chem. Soc.* 139, 2070–2082 (2017).

3) Görlin, M. et al. Formation of Unexpectedly Active Ni-Fe Oxygen Evolution Electrocatalysts by Physically Mixing Ni and Fe Oxyhydroxides. *Chem. Commun.* (2018). doi:10.1039/C8CC06410E

Work concerning in situ XAS - CV cycling of Ni-Fe(OxHy) catalysts:

4) González-Flores, D. et al. Nickel-iron catalysts for electrochemical water oxidation – redox synergism investigated by in situ X-ray spectroscopy with millisecond time resolution. *Sustain. Energy Fuels* 2, 1986–1994 (2018).

5) Drevon, D. et al. Uncovering The Role of Oxygen in Ni-Fe(OxHy) Electrocatalysts using In situ Soft X-ray Absorption Spectroscopy during the Oxygen Evolution Reaction. *Sci. Rep.* 9, 1532 (2019).

Comment #2

In the UV-vis-CV cycling experiments by Smith et al. (ref. 30 in main manuscript), the CV curves show an indication of two distinct metal oxidation processes (pre-catalytic and catalytic) separated by a plateau. This plateau is not recognized in the other in situ UV-vis studies of Ni-Fe catalysts listed in Comment #1. For clarity and comparison to these studies, it would be desirable if you presented the UV-vis traces (at a fixed wavelength or region of interest) during CV cycling of your catalysts. It could add clarity to whether it is possible to recognize and discern the [+] and [++] states in your Ni-Fe catalysts.

Comment #3

In Figure S6, how do the "catalytic" UV-vis spectra look when you have subtracted the spectrum at OCP (0.237 V overpotential) as background instead of the spectrum at 0.307 V overpotential? Please make a comparison of these two spectra.

Comment #4

You provide nice and detailed experiments on the [+] and [++] states. You also carry out catalytic potential-OCP switching experiments where the rate of the spontaneous relaxation of the oxidized state without the impact of a potential driving force (water oxidation kinetics). First, how do you define this process relaxation process and how does it relate to water oxidation kinetics? Second, in Figure S8-S10 where you carry out potential switching to and from fixed potentials, you do not attempt to estimate the rate of oxidation/reduction. It would be desirable to fit these time traces of the oxidative jump (0.24 – 0.54 V overpotential) and the reductive jump (0.54 V to 0.24 V overpotential) in order to obtain the rate constants of the [++] oxidation state. (In addition, you could perform the same potential jump experiments from 0 to 0.24 V overpotential to obtain the oxidation rate of the [+] state.) This would be helpful in the discussion of the kinetics of the [+] and [++] states in addition to the water oxidation kinetics.

Comment #5

In the main manuscript, page 6, line 3-5. "We note that FeOOH(++) states are not generated in non-aqueous electrolytes, highlighting the important role of water molecules and/or proton-coupled electron transfer in the generation of these states (see Figure S11c in ESI)". You could point out that the oxidation process of FeOOH (hidden in the OER current in water-based electrolytes) can be identified in nonaqueous electrolytes with a small addition of water. Also, please convert the x-axis in Figure S11 to overpotential (not Ag/AgCl) for consistency.

Comment #6

Concerning the notation "singly" [+] and "doubly" [++] oxidized states. Usually, UV-vis is very uncertain in determining the absolute metal oxidation states, where another complementary X-ray method is desired. Also, I notice that the increase of the [+] and [++] states in your UV-vis data do not exceed 100 % (i.e. do not exceed +1 unit in relative oxidation state). This can be determined from Figure S2 and the metal loadings provided from the ICP-OES listed in Table S1. The Ni(Fe)OOH catalyst is assumed to have both the [+] and [++] states Ni-centred, whereas the FeOOHNiOOH and FeOOH catalysts have the [+] state Ni-centred and the [++] state Fe-centred (as you discuss in your data). Assuming that every oxidized state originates from the loss of one electron, the numbers of oxidized states in Figure 2 can be converted into moles of oxidized states (mol/cm²) using Avogadro's number. So, after dividing the moles of [++] states with the moles of atoms on the electrode from ICP-OES, the percentages of [+] states in the Ni(Fe)OOH, FeOOHNiOOH, and FeOOH catalysts are estimated to 83 %, 8 %, and 2 %, respectively. The estimated [++] states are 9 %, 9 %, and 4 %, respectively. The total sum of the [+] and [++] states are therefore 92 %, 17 %, 6 %, respectively. This shows that none of the metal sites increases more than +1 in oxidation state. I am aware of the possibility that only edge- or defect sites participate in OER [see reference 6, Burke Stevens, below]. However, since both UV-vis spectroscopy is a bulk method, it should reflect the average oxidation state of the atoms in the film, were each metal site never exceeds +1 unit. My point is that the

notation of [+] vs [++] states should be used with care, and you need to be more careful with defining these in the text so there are no confusions with absolute oxidation states, unless you can confirm the oxidation state definitely of the metal sites with a complementary X-ray technique.

6) Burke Stevens, M., Trang, C. D. M., Enman, L. J., Deng, J. & Boettcher, S. W. Reactive Fe-Sites in Ni/Fe (Oxy)hydroxide Are Responsible for Exceptional Oxygen Electrocatalysis Activity. *J. Am. Chem. Soc.* 139, 11361–11364 (2017).

Comment #7

Are the dat curves in Figure S15 the same data as presented in Figure S12-13? In Figures S13 and S15, please provide the potential prior to switching off to OCP in the figure caption. It will be easier to follow.

Comment #8

In Figure S16 you state that the measurements were performed in pH 7. Please specify which electrolyte was used for these measurements. Please also provide the details in the Experimental section.

Comment #9

In the SEM cross sections in Figure S2, it looks like a hydroxide layer is precipitated on the electrode without passing of charge (the film with "0 mC/cm²")? Is this correct?

Comment #10

According to the given metal loadings in the Ni(Fe)OOH, FeOOHNiOOH, and FeOOH catalysts given from the ICP-OES listed in Table S1, the differences in loadings on the electrodes are quite large. This however is not reflected in a similar difference in film thicknesses in the SEM cross-sections presented in Figure S1-S2. Please comment on this discrepancy.

Other comments

- In Table 1, please check that it is correct that the information in entry 10 (τ (s)) was obtained from Figure S11.
- In Figure 2c, check the right Y-axis, whether it applies only to the [++] state, or also to the [+] state?
- In Figure S5, (a) and (b) labels are missing.
- In Figure S7, please provide a legend for all colored curves.
- In Figures S8-10, please specify what the different colors represent in the figure caption and/or provide a legend for the coloured curves.

Reviewer #2 (Remarks to the Author):

In this paper the authors used operando spectroelectrochemistry to study FeOOH and two samples of NiFeOOH-type catalysts. These catalysts are widely studied due to relevance to water splitting. The active site of these catalysts and the mechanism are still under debate. The authors provide unique data and are able to optically characterized presumed intermediates in different catalysts. These data together with their analysis are important to the field. Overall the paper has a high technical quality. In principle the paper is suitable for Nat Commun. However, I feel there are some important technical issues to address. Also the mechanistic hypothesis seems to have gone too far given their data. With a major revision in these aspects, the paper's quality will be improved.

1. I know it is hard to collect lots of data. But I cannot help wondering why the authors did not study

NiOOH, that is without any Fe. This is important given the authors' argument that in Ni(Fe)OOH the oxidation is Ni centered. If there is a reason not to do it, then the authors should revise their discussion on this part. In fact here they say something about Ni-O-Fe, then Ni centered. For me this is not clear. Do you mean the oxidation can be delocalized so the absorption is different from a pure FeOOH sample? That is more reasonable and clear than saying Ni centered.

2. NiOOHFeOOH. I am not sure what this sample is. Are you sure they are phase separated? You show pre-catalyst separation. But how about after activation? The work of Bell et al. (JACS 2015) showed that at high Fe level, one would make NiFeOx plus FeOOH. Are you sure you are not making it? The paper emphasizes the "interface" between NiOOH and FeOOH. But one need to make sure this phase separation dominates.

3. Fig. 1. Please note the loading in the caption. The geometric activity seems much lower than typical NiFeOx, but it might be due to a low loading. Please clarify.

4. When you report experiments at 2 ma/cm², please also report the potential. It looks that the authors largely neglect potential-dependent activity in this paper. See also below. I feel this might be overlooked - rate can be potential dependent in addition to concentration of resting/intermediates.

5. The authors make important assumptions in the estimation of concentrations for oxidized and doubly oxidized species. I think it is better to explain the assumption in the main text.

6. The kinetic analysis is an important contribution to the field. However, I feel the current analysis need to clarify a few points. (a) As the authors are aware of, UV-Vis spectroscopy can only catch species with distinguished optical absorption. Differential spectroscopy only detects obvious differences. Is it possible that the real intermediates cannot be seen? (b) You assume one electron for each species you see. What is the proof? In literature people tend to assign 1.7 for first oxidation in Ni. In any case, please discuss these issues. (c) More importantly, according to your analysis, you are not seeing the resting states, right? what is the percentage of doubly oxidized species in your film? This is important for later mechanistic analysis. (d) Fig. 4a neglects potential influence. Justified? An electrochemical reaction will depend somehow to potential in terms of rate, unless a very special condition applies. Please clarify and discuss. You study the decay off potential. Is this rate exactly match the current density? In other words, are you suggesting that once the doubly oxidized species is made, all reactions are pure chemical in nature? (e) How do you explain the difference of 4a and 4b in terms of order of TOF? If potential is not important, the trends should be the same, no?

7. Please add TOF at 300 mV in Table 1.

8. In my opinion the section on Mechanistic discussion lacks solid support and limits the impact of the experimental work. As the authors acknowledge, they see only an intermediate before the RDS. They cannot probe the chemical nature. Moreover, they can only follow the decay of this species, while not being able to track proton transfer etc. Their discussion also does not reconcile with literature data in electrokinetics. They say "this disparity is can be attributed to the non-ideal nature of charge accumulation in these materials, with the different Tafel slopes most likely originating from differences in the dependence of charge density upon applied potential (Figure 2d), rather than being indicative of different reaction mechanisms." What does it mean non-ideal nature of charge accumulation? I have no idea. Keeping in mind Tafel studies include terms on potential dependence, while the data shown by the authors don't. I agree simple Tafel analysis is not easy to give a mechanism, but the contradiction with Tafel data is not easily dismissed.

9. The mechanism plots in Fig 5 are actually wrong. You should have only 4 e for OER, not 0 e as drawn. The addition of 4 e in RDS is strange. In any case, the data shown in the paper don't tell much about the mechanism. I recommend to mostly remove this section except keeping the discussion of "the nature of states" paragraph.

Reviewer #3 (Remarks to the Author):

The authors report a kinetic study of the water oxidation catalysis by nickel-iron oxyhydroxides. The

main conclusions are that four equivalents of oxidized species are required for turnover to occur, that the kinetics have a fourth-order dependence on oxidized species, and that iron in itself is not responsible for the observed oxidized species. This is a well-performed study which complements the considerable data available on the iron-nickel system and is particularly useful in that it compares FeOOH and iron-rich mixed oxyhydroxide materials. Ultimately, however, the conclusions drawn are tenuously supported by the data. The paper would be more useful if it presented the data with all likely interpretations, rather than making a case that isn't there. There are a few peculiar findings of the study which require a more detailed treatment prior to publication, and I think that the conclusions relating to the role of iron centers is overstated and unsupported generally.

To begin with, the authors assign the absorption features in Figure 2a,b to d-d interband transitions and cites reference 27. If you look at reference 27, that assignment is just a throwaway - there is no justification for the assignment. Metal oxides and hydroxides like this tend to be insulators because, even though the d-orbitals aren't filled, they don't lead to conduction because the orbitals tend to be strongly localized at the metal centers; not delocalized into bands. So, the absorption features shown in Figure 2 are unlikely to be d-d absorptions, especially because of their intensities. It seems charge transfer is more likely. Because of this, it is difficult to draw the conclusion that any of the features are centered on a particular metal, which the authors use to strongly implicate nickel. I don't see this as a compelling argument.

The Ni(Fe)OOH spectrum in Figure 2b is pretty interesting. Durrant in an earlier paper (Advances in Photoelectrochemical Water Splitting: Theory, Experiment and Systems Analysis, Chapter 5, available online) attributed broad 600 nm absorption features to Fe(IV)=O chromophores. This is in line with other studies which detect high-valent iron, but the authors do not consider this. Why?

The bar graph for Ni(Fe)OOH in Figure 2c would also be consistent with charge removal from Fe in the second (+/++) step, given its reduced amplitude relative to the charge removal at (0/+). This trend is not seen for the other materials, and is an important distinction.

On p.12, the authors claim that the spectrum of Ni(Fe)OOH(+++) doesn't change as a function of applied potential, and says this is an unlikely observation if the accumulation of 4 oxidative equivalents occurred on single isolated Fe centers distributed near the edges of the material. First, this is the appropriate place to cite and discuss Hunter et al. (Joule), and second, the authors need to explain why it is unlikely; I don't see this.

The fourth order rate law is a little curious as well. First, fourth order reactions do not happen in elementary steps. Second, I suspect that the rapid decay of the Ni(Fe)OOH(+++) absorbance when the applied potential is removed, which is attributed to water oxidation, does not follow fourth-order kinetics. A fourth order reaction is protracted over a very long time period (compared to a first order reaction of comparable time constant). If the water oxidation rate depended on the fourth power of [Ni(Fe)OOH(+++)], then it would be very clear from the decays in S12 and S14. The kinetic fits in these figures is not convincing. Moreover, a plot of $1/[\text{Ni(Fe)OOH(+++)}]^3$ should be linear. It is likely that the [Ni(Fe)OOH(+++)] decay kinetics are not consistent with a fourth order reaction and, contrary to what the authors say, are not consistent with Figure 4a. What they measure in Figure 4a, is a steady state concentration of [Ni(Fe)OOH(+++)] as a function of applied potential; their interpretation is the simple rate law in equation 1. But equation 1 embodies many elementary steps, and is unlikely to be consistent with the data in Figures S12 and S14.

Based on the fact that the kinetics are not well resolved, some of the claims in the paper are dubious. On page 4, the authors note "[species] accumulated in Ni(Fe)OOH are distinct, with greater Ni character." While it is clear that the features are distinct, what is the justification for claiming nickel character? This material is fundamentally different than the other materials synthesized, so why would sequential iron oxidation in this material look similar across all three? If these bands are indeed charge transfer bands, which is more likely than d-d transitions, wouldn't the local coordination

environment of iron and nickel be the most pertinent difference? Similarly, the claim on page 8 line 13 that the states "probably [have] significant Ni character" is not supported by the discussion above in this light. This argument persists through the paper, concluding with page 12 line 18: "This would indicate that these... states are centered on Ni atoms, instead of on Fe." A more robust analysis is required for these claims, since they do not seem to be uniquely supported by the data.

An alternative conclusion from the data presented on page 9, lines 3-15 is that the Ni(Fe)OOH has fewer iron sites, resulting in less formation of the double oxidized species.

I am not convinced that the difference in Tafel slopes indicates the "non-ideal nature of charge accumulation" rather than a change in mechanism. Surely differences in charge accumulation will play a role in the Tafel slope, but in general vastly different Tafel slopes are not entirely attributable to this, rather to different underlying mechanisms.

The initial "charging" current needed to bring the catalyst to its resting state (generally agreed to be Ni(III)/Fe(III)) is well documented, so the discussion on page 7 (lines 1-8) should be tied in to the existing literature, rather than presented as a new discovery.

There is a minor typographical error on page 6, line 5. The superscript on cm should be changed from 2 to -2.

Response to Reviewers Comments

Reviewer 1 (Remarks to the Author):

Summary:

The manuscript deals with UV-vis spectroscopic investigations of electrodeposited Ni-Fe(OxHy) catalysts, currently the most promising catalyst for the alkaline water oxidation reaction (OER). The nature of the metal redox states during applied catalytic potential is still under debate owing to a complex redox cycle that extends into the OER regime. This manuscript provides new insight into this complexity of the redox system by utilizing time-resolved in situ UV-vis spectroscopy. The experiments are well performed and the results reveal new surprising insights, where two spectroscopic fingerprints are discerned; both Ni-centred oxidation (the [·] state) prior to OER and Fe-centred metal oxidation hidden in the catalytic regime (the [··] state). Further kinetics are resolved by evaluating kinetic time traces, and information is obtained regarding water oxidation kinetics. However, there are some question marks that need to be thoroughly addressed, and a few already published work on time-resolved UV-vis spectroscopy (and time-resolved X-ray absorption spectroscopy) on Ni-Fe(OxHy) catalysts has not been recognized in the current manuscript.

I would recommend this manuscript for publication since it provides a new detailed and unique set of data along with a clarification of metal-centred redox states in Ni-Fe catalysts. However, I would recommend the authors to carefully address the specific concerns before a final decision is reached.

We thank Reviewer 1 for their summary of our work and for making us aware of some recent work that we may have overlooked in our manuscript. We have now considered these publications and made reference to them where suitable in our manuscript, as detailed below.

Comment #1

Make sure that relevant literature on the topic is recognized and cited in the manuscript. The articles listed below deal with “dynamic” and time-resolved in situ UV-vis or X-ray absorption spectroscopic measurements of Ni-Fe(OxHy) catalysts. Rates for the metal oxidation/reduction process are discussed in the in situ UV-vis study by Loos et al. [1], both during CV-cycling and similar potential jumps as presented in your study. The rate constants of the oxidation processes from in situ XAS data are presented in the work of González-Flores et al. [4] which includes both CV cycling and potential jumps. In the work of Görlin et al. [2-3] in situ UV-vis data is presented during CV-cycling of both Ni-Fe catalysts and of physically mixed Ni+Fe catalysts.

We are pleased to have been made aware of recent literature that we were not previously aware of. We have added reference to these in our manuscript and believe that they strengthen our conclusions. Please see the details below.

Work on in situ UV-vis studies of Ni-Fe(OxHy) catalysts:

- 1) Loos, S., Zaharieva, I., Chernev, P., Lißner, A. & Dau, H. Electromodified NiFe Alloys as Electrocatalysts for Water Oxidation: Mechanistic Implications of Time-Resolved UV/Vis Tracking of Oxidation State Changes. ChemSusChem 0, (2019)

Added (reference 53) and referred to on page 13, lines 27-29:

“¹⁸O-labeling experiments by Hu et al. also indicate different active sites for Ni and NiFe oxides (Fe ≈ 25%),⁵² as do light scattering experiments on similar samples by Dau et al., in-line with our conclusions herein.⁵³”

- 2) Görlin, M. et al. Tracking Catalyst Redox States and Reaction Dynamics in Ni–Fe Oxyhydroxide Oxygen Evolution Reaction Electrocatalysts: The Role of Catalyst Support and Electrolyte pH. *J. Am. Chem. Soc.* 139, 2070–2082 (2017).

This article was cited in our original version of the manuscript (reference 27) in relation to colour changes of NiOOH. Additional mention in the revised manuscript was made (page 4, lines 16-19):

“For Ni(Fe)OOH, the first (0/+) oxidation has been assigned to the oxidation of Ni(OH)₂ to NiOOH as observed by others,²⁷ with the associated optical absorption increase assigned to nickel d-d interband transitions,²⁸ although the details of this assignment are not critical for the study herein.”

- 3) Görlin, M. et al. Formation of Unexpectedly Active Ni-Fe Oxygen Evolution Electrocatalysts by Physically Mixing Ni and Fe Oxyhydroxides. *Chem. Commun.* (2018). doi:10.1039/C8CC06410E

The above citation has been added to our revised manuscript (reference 54) (page 13, lines 31-32):

“It also cannot be ascertained from these data alone whether a regime exists in which both metals are responsible for charge accumulation, as reported by others.^{31,41,54”}

Work concerning in situ XAS - CV cycling of Ni-Fe(OxHy) catalysts:

- 4) González-Flores, D. et al. Nickel-iron catalysts for electrochemical water oxidation – redox synergism investigated by in situ X-ray spectroscopy with millisecond time resolution. *Sustain. Energy Fuels* 2, 1986–1994 (2018).

The above citation has been added to our revised manuscript (reference 45) (page 8, lines 6-10):

“Overall, these data confirm that the (0/+) process observed in our spectroelectrochemical data is an oxidation process which, though important for catalytic function, is not directly involved in water oxidation catalysis. This is in accordance with other documentation in the literature for such mixed metal oxides reporting an ‘activation’ or pre-catalytic step to bring the electrode to the catalytic resting state.^{27,45”}

- 5) Drevon, D. et al. Uncovering The Role of Oxygen in Ni-Fe(OxHy) Electrocatalysts using In situ Soft X-ray Absorption Spectroscopy during the Oxygen Evolution Reaction. *Sci. Rep.* 9, 1532 (2019).

This article is now cited (reference 49) in our revised manuscript (page 13, lines 6-8):

“This high order of reaction would also agree with reports suggesting that the RDS is related with oxygen release,²⁷ and the observation of accumulated oxidised species in XAS experiments prior to O-O bon formation.^{49”}

Comment #2

In the UV-vis-CV cycling experiments by Smith et al. (ref. 30 in main manuscript), the CV curves show an indication of two distinct metal oxidation processes (pre-catalytic and catalytic) separated by a plateau. This plateau is not recognized in the other in situ UV-vis studies of Ni-Fe catalysts listed in Comment #1. For clarity and comparison to these studies, it would be desirable if you presented the UV-vis traces (at a fixed wavelength or region of interest) during CV cycling of your catalysts. It could add clarity to whether it is possible to recognize and discern the [+] and [++] states in your Ni-Fe catalysts.

At the reviewers requested, we have added the data below to our revised manuscript in the supporting information (Figure S14). We have performed this experiment for Ni(Fe)OOH, monitoring the absorbance at 650 nm. While no distinct plateau is readily visible in the optical data between the redox and catalytic currents, a change in the absorption profile may be noted in moving between the two regimes. Furthermore, the absorption increase is greater for the pre-catalytic species, [+], than the species accumulated over catalytic potentials, [++], despite the wavelength better suited to tracking of the latter, thereby reflecting the difference in the number of these species, as shown in Figure 2c in the manuscript, as well as highlighting the different extinction coefficients for the [+] and [++] species. The following addition has been added to the revised manuscript (page 8, lines 4-6):

“Similar data and conclusions were obtained for the other two electrocatalysts, including UV-Vis-CV cycling (see ESI, Figures S14 and S15).”

Figure S14: Simultaneous measurement of the optical signal at 650 nm (teal) and current (burgundy) during CV cycling of a sample of Ni(Fe)OOH at 1 mV/s in 0.1 M NaOH. At the ca. the catalytic onset, the absorbance profile changes shape, reflecting a shift from the accumulation of [+] to [++] species.

Comment #3

In Figure S6, how do the “catalytic” UV-vis spectra look when you have subtracted the spectrum at OCP (0.237 V overpotential) as background instead of the spectrum at 0.307 V overpotential? Please make a comparison of these two spectra.

The spectra given in Figure S6 show the ‘pre-catalytic’ species spectra (calculated by subtracting the initial absorption from the absorption after activation). This activation raises the OCP. The spectra of the ‘catalytic’ species is then calculated by subtracting the absorption spectra at this new OCP from the spectra generated at higher potentials. Besides the change in OCP, it is also very difficult to make the comparison requested, as the greater extinction coefficient for the [·] species means that it dominates the UV-Vis spectra.

Comment #4

You provide nice and detailed experiments on the [·] and [··] states. You also carry out catalytic potential-OCP switching experiments where the rate of the spontaneous relaxation of the oxidized state without the impact of a potential driving force (water oxidation kinetics). First, how do you define this process relaxation process and how does it relate to water oxidation kinetics? Second, in Figure S8-S10 where you carry out potential switching to and from fixed potentials, you do not attempt to estimate the rate of oxidation/reduction. It would be desirable to fit these time traces of the oxidative jump (0.24 – 0.54 V overpotential) and the reductive jump (0.54 V to 0.24 V overpotential) in order to obtain the rate constants of the [··] oxidation state. (In addition, you could perform the same potential jump experiments from 0 to 0.24 V overpotential to obtain the oxidation rate of the [·] state.) This would be helpful in the discussion of the kinetics of the [·] and [··] states in addition to the water oxidation kinetics.

Firstly, we define the initial relaxation after switching to OCP as the rapid consumption of [··] states accumulated at the surface due to water oxidation. This initial decay, shown close-up in Figure S15 in the revised manuscript, agrees with the rate of water oxidation that we obtained from our step-potential spectroelectrochemistry analysis. Indeed we do fit these decays, the rate constants for which are given in Figure S18. Unfortunately, we are not able to do the same experiment over the pre-catalytic potential window as the sample is not particularly stable at more negative (reductive) potentials.

Comment #5

In the main manuscript, page 6, line 3-5. “We note that FeOOH(··) states are not generated in non-aqueous electrolytes, highlighting the important role of water molecules and/or proton-coupled electron transfer in the generation of these states (see Figure S11c in ESI)”.

You could point out that the oxidation process of FeOOH (hidden in the OER current in water-based electrolytes) can be identified in nonaqueous electrolytes with a small addition of water. Also, please convert the x-axis in Figure S11 to overpotential (not Ag/AgCl) for consistency.

Thank you for the suggestion. This result is not particularly strong as the current is very small, but we have now made mention of this in the following lines (page 7, lines 4-6):

“Upon addition of water, a very small oxidative wave can be determined, normally masked by the OER current in aqueously electrolytes, reaffirming the localised, ‘molecular’ style oxidations in these films.”

Unfortunately, we cannot readily convert the x-axis into overpotential because the electrochemistry was not performed in aqueous electrolyte. We therefore used the reference electrode as the reference marker.

Comment #6

Concerning the notation “singly” [+] and “doubly” [++] oxidized states. Usually, UV-vis is very uncertain in determining the absolute metal oxidation states, where another complementary X-ray method is desired. Also, I notice that the increase of the [+] and [++] states in your UV-vis data do not exceed 100 % (i.e. do not exceed +1 unit in relative oxidation state). This can be determined from Figure S2 and the metal loadings provided from the ICP-OES listed in Table S1. The Ni(Fe)OOH catalyst is assumed to have both the [+] and [++] states Ni-centred, whereas the FeOOHNiOOH and FeOOH catalysts have the [+] state Ni-centred and the [++] state Fe-centred (as you discuss in your data). Assuming that every oxidized state originates from the loss of one electron, the numbers of oxidized states in Figure 2 can be converted into moles of oxidized states (mol/cm²) using Avogadro’s number. So, after dividing the moles of [++] states with the moles of atoms on the electrode from ICP-OES, the percentages of [+] states in the Ni(Fe)OOH, FeOOHNiOOH, and FeOOH catalysts are estimated to 83 %, 8 %, and 2 %, respectively. The estimated [++] states are 9 %, 9 %, and 4 %, respectively. The total sum of the [+] and [++] states are therefore 92 %, 17 %, 6 %, respectively. This shows that none of the metal sites increases more than +1 in oxidation state. I am aware of the possibility that only edge- or defect sites participate in OER [see reference 6, Burke Stevens, below]. However, since both UV-vis spectroscopy is a bulk method, it should reflect the average oxidation state of the atoms in the film, were each metal site never exceeds +1 unit. My point is that the notation of [+] vs [++] states should be used with care, and you need to be more careful with defining these in the text so there are no confusions with absolute oxidation states, unless you can confirm the oxidation state definitely of the metal sites with a complementary X-ray technique.

Indeed, we agree with the Reviewer that UV-vis spectroscopy alone is not adequate to determine absolute oxidation states, and we had never intended to suggest oxidation states with our nomenclature. As mentioned in the text, [+] and [++] are intended to denote the states formed following the first and second oxidation steps that we monitor with our techniques. We found this notation to be the easiest for readers to follow, but to aid understanding, we have added to following clarification to the revised manuscript (page 4, lines 11-14):

“Note that this notation is only intended to describe an increased oxidation state and does not give any indication of the absolute oxidation state of the metal centres, which could only be obtained with a suite of complementary techniques beyond the scope of this study.”

Regarding presentation of the data, please note that the moles of oxidised states are provided in the second y-axis of Figure 2 in the original and revised manuscript. The percentages calculated by the reviewer represent the idea that, if all metal centres were active catalytic sites, none would be sufficiently oxidised to perform catalysis.

But, as mentioned with the citation provided by the reviewer, it is far more probable that only a few of the metal atoms are active, consistent with a recent study that demonstrated that only the outer ~1 nm of NiFe_xO_yH_z particles are redox active (Roy, C. et al, Nature Catalysis 2018). UV-Vis spectroscopy is a bulk technique, but spectroelectrochemistry will only give the spectral changes for those metal centres that change oxidation state, thus the data presented in the manuscript does not reflect the average oxidation states of the atoms in the film.

In line with the percentages calculated by the reviewer, we have added the following to the supporting information (page 21):

“The quantity of [+] and [++] species can be calculated as a percentage of metal total number of metal centres from the ICP-OES analysis given in Table S1. This calculation is given for the [+] species in Section E and below for the [++] species. The results are summarised in Table S2.

[+]

$$\% \text{FeOOH}[+] \rightarrow \text{mol of Fe}[+] (\text{SEC, Figure 2c}) / \text{mol of Fe (ICP, Table S1)} = 1.4 \times 10^{-9} / 2.4 \times 10^{-7} = 0.005 \rightarrow \sim 0.5\%$$

$$\% \text{FeOOHNiOOH}[+] \rightarrow \text{mol of Ni}[+] (\text{SEC, Figure 2c}) / \text{mol of Ni (ICP, Table S1)} = 5.7 \times 10^{-9} / 6.22 \times 10^{-8} = 0.09 \rightarrow \sim 9\%$$

$$\% \text{Ni(Fe)OOH}[+] \rightarrow \text{mol of Ni}[+] (\text{SEC, Figure 2c}) / \text{mol of Ni (ICP, Table S1)} = 3.9 \times 10^{-8} / 4.63 \times 10^{-8} = 0.85 \rightarrow \sim 85\%$$

[++]

$$\% \text{FeOOH}[++] \rightarrow \text{mol of Fe}[++] (\text{SEC, Figure 2c}) / \text{mol of Fe (ICP, Table S1)} = 9.8 \times 10^{-9} / 2.4 \times 10^{-7} = 0.04 \rightarrow \sim 4\%$$

$$\% \text{FeOOHNiOOH}[++] \rightarrow \text{mol of Fe}[++] (\text{SEC, Figure 2c}) / \text{mol of Fe (ICP, Table S1)} = 2.2 \times 10^{-8} / 2.4 \times 10^{-7} = 0.09 \rightarrow \sim 9\%$$

$$\% \text{Ni(Fe)OOH}[++] \rightarrow \text{mol of Ni}[++] (\text{SEC, Figure 2c}) / \text{mol of Ni (ICP, Table S1)} = 5.3 \times 10^{-9} / 4.6 \times 10^{-8} = 0.12 \rightarrow \sim 12\%$$

Table S2

Samples	% [+] species per total M-centres*	% [++] species per total M-centres*
FeOOH	0.5	4
FeOOHNiOOH	9	9
Ni(Fe)OOH	85	12

**where ‘M’ is the active metal MOOH(++) species for each sample type.”*

We have also added this short line to the revised manuscript (page 5, lines 10-11):

“The quantity of each MOOH(+) and MOOH(++) species is also provided as a percentage of the number of corresponding M metals (Table S2).”

6) Burke Stevens, M., Trang, C. D. M., Enman, L. J., Deng, J. & Boettcher, S. W. Reactive Fe-Sites in Ni/Fe (Oxy)hydroxide Are Responsible for Exceptional Oxygen Electrocatalysis Activity. *J. Am. Chem. Soc.* 139, 11361–11364 (2017).

Comment #7

Are the dat curves in Figure S15 the same data as presented in Figure S12-13? In Figures S13 and S15, please provide the potential prior to switching off to OCP in the figure caption. It will be easier to follow.

Our apologies if the data was unclear. The data presented in Figure S18 (previously S15) is the same as that given in Figures S15 and S16 (previously S12-13). The potential applied, prior to switching to OCP, was 1.7 V vs RHE. This has been added to the caption for these Figures.

Comment #8

In Figure S16 you state that the measurements were performed in pH 7. Please specify which electrolyte was used for these measurements. Please also provide the details in the Experimental section.

The electrolyte used was potassium phosphate buffer. The following details have been added to the experimental section (page 3, section B):

“A 0.1 M sodium hydroxide (pH 13) was used as the electrolyte in all experiments, other than the analysis in Figure S19, which used 0.1 M potassium phosphate buffer (pH 7).”

Comment #9

In the SEM cross sections in Figure S2, it looks like a hydroxide layer is precipitated on the electrode without passing of charge (the film with "0 mC/cm²")? Is this correct?

Figure S2 only concerns the deposition of NiOOH on top of FeOOH to form FeOOHNiOOH. As such, the hydroxide layer visible at 0 mC/cm² is the FeOOH that was previously deposited (80 nm thick). This thickness does not change during the NiOOH deposition until 100 mC/cm² of charge is passed, indicating that, until this value is exceeded, the NiOOH phase is depositing within the pores of the underlying FeOOH. To aid clarity, the Figure S2 caption has been altered to read as follows:

“SEM images for the deposition of NiOOH on top of pre-deposited FeOOH to form FeOOHNiOOH: Top view (top) and side view (bottom) images of FeOOHNiOOH films taken after passing 0 mC/cm², 50 mC/cm², 100 mC/cm², and 150 mC/cm² for NiOOH deposition.”

Comment #10

According to the given metal loadings in the Ni(Fe)OOH, FeOOHNiOOH, and FeOOH catalysts given from the ICP-OES listed in Table S1, the differences in loadings on the electrodes are quite large. This however is not reflected in a similar difference in film thicknesses in the SEM cross-sections presented in Figure S1-S2. Please comment on this discrepancy.

The discrepancy is explained in ESI (Section A, 3rd paragraph):

“When the FeOOH and NiOOH films were prepared with comparable thicknesses, ICP-OES results showed that the amount of Ni present in the NiOOH film was approximately 1/5 the amount of Fe present in the FeOOH film (Table S1). This is because NiOOH is composed of vertically oriented, loosely packed, larger sheets while FeOOH is composed of more densely packed smaller plates (Figure S1).”

Other comments

- In Table 1, please check that it is correct that the information in entry 10 (τ (s)) was obtained from Figure S11.

Our apologies for the incorrect labelling. The Table 1, entry 11 data (previously entry 10) comes from Figure 3b and Figure S15. This has been corrected in the revised manuscript.

- In Figure 2c, check the right Y-axis, whether it applies only to the [++] state, or also to the [+] state?

The right y-axis in Figure 2c was meant to refer to both states and the axis title has been adjusted accordingly to reflect this.

- In Figure S5, (a) and (b) labels are missing.

This has been corrected in the revised manuscript

- In Figure S7, please provide a legend for all coloured curves.

At the Reviewer's request, we have added a legend to show exactly which potential each colour represents.

- In Figures S8-10, please specify what the different colours represent in the figure caption and/or provide a legend for the coloured curves.

A legend has been added to Figure S10a, S11a and S12a to detail the potentials that were applied.

Reviewer #2 (Remarks to the Author):

In this paper the authors used operando spectroelectrochemistry to study FeOOH and two samples of NiFeOOH-type catalysts. These catalysts are widely studied due to relevance to water splitting. The active site of these catalysts and the mechanism are still under debate. The authors provide unique data and are able to optically characterized presumed intermediates in different catalysts. These data together with their analysis are important to the field. Overall the paper has a high technical quality. In principle the paper is suitable for Nat Commun. However, I feel there are some important technical issues to address. Also the mechanistic hypothesis seems to have gone too far given their data. With a major revision in these aspects, the paper's quality will be improved.

We hope that with the following revisions to the manuscript, we are now able to convince Reviewer 2 of the strength of our hypothesis regarding reactive centres and mechanism.

1. I know it is hard to collect lots of data. But I cannot help wondering why the authors did not study NiOOH, that is without any Fe. This is important given the authors' argument that in Ni(Fe)OOH the oxidation is Ni centered. If there is a reason not to do it, then the authors should revise their discussion on this part. In fact here they say something about Ni-O-Fe, then Ni centered. For me this is not clear. Do you mean the oxidation can be delocalized so the absorption is different from a pure FeOOH sample? That is more reasonable and clear than saying Ni centered.

Following the advice we have received from Reviewer 2, we have now included new data on NiOOH that was synthesised and tested in electrolyte free from any Fe impurities. We note that, as shown in new Figure S8, that the spectral feature of the [++] species accumulated closely resembles that of the Ni(Fe)OOH sample discussed in the rest of the manuscript. As such, we can conclude that this peak must result from charge accumulation on Ni centres, the only metal common to both samples. We had previously believed, in accordance with literature, that the difference in absorption for the Ni(Fe)OOH sample compared to the other Fe-dominant samples, indicated a degree of delocalisation of the [++] species towards to the Ni centre in any Ni-O-Fe bonds, as mentioned by the reviewer. However, we have now modified our manuscript to emphasise that the species accumulated in the steady state of catalysis must be largely centred on the Ni atoms in Ni(Fe)OOH. We believe that the addition of this data greatly strengthens our manuscript and will be of significant interest to the field. To express this, the following edits have been made to the revised manuscript:

(page 4, lines 32-47)

“This suggests that the MOOH(++) species accumulated in FeOOH and FeOOHNiOOH during water oxidation catalysis are similar in nature (and therefore both Fe-centred), whilst those accumulated in Ni(Fe)OOH are distinct, likely alluding to greater Ni character. To better determine whether this is the case, a NiOOH sample free of any Fe impurities was prepared and analyzed in a purified electrolyte and otherwise comparable conditions (see ESI, Section F for details). As shown in Figure S8, the same two spectral features present for Ni(Fe)OOH are observed for the purified NiOOH sample: a (0/+) oxidation, assigned to the oxidation of any Ni(OH)₂; and a second

(+/++) oxidation process, which gives rise to a peak at ca. 650 nm. As such, we can propose that MOOH(++) species accumulated under catalytic conditions in these mixed metal systems are Fe-centred at higher concentrations of iron, but Ni-centred at low Fe concentrations ($\leq 5\%$). It may be the case that there are intermediate Ni/Fe ratios for which either both metal centres can accumulate charge. It is also possible that this charge becomes delocalized between centres, which would agree with spectra recently reported by Dau and coworkers for mixed Ni/Fe electrocatalysts, assigned to di- μ -oxo bridged Ni(IV)-Fe(III) motifs.³² Hu and coworkers also propose dual-site water oxidation pathways with certain Ni/Fe oxyhydroxide structures,³⁰ and a synergistic mechanism remains probable given the improved catalytic onset of FeOOHNiOOH compared to FeOOH, as discussed below.”

(page 9, lines 12-14)

“On the other hand, our values are lower than those reported on Ni(Fe)OOH per Fe atom, suggesting that the Fe concentration in our samples is lower than the measured operando MOOH(++) state density, and as such, reaffirms that Ni(Fe)OOH(++) are Ni-centred, as discussed above.”

(page 10, lines 10-15)

“The faster water oxidation kinetics of Ni(Fe)OOH in conjunction with the accumulation of fewer Ni(Fe)OOH(++) species results in a similar overall performance to FeOOHNiOOH. These differences in reactivity between FeOOHNiOOH and Ni(Fe)OOH suggest a different nature of the MOOH(++) species, in agreement with the different spectral features, attributed to Fe- vs Ni-centred oxidized species, respectively. The overall result is then a comparable J-V curve for both samples.”

(page 13, lines 17-35)

“These spectra suggest that Ni(Fe)OOH(++) states are associated either with oxidation of sparsely incorporated Fe within a NiOOH environment, or with Ni groups. Given the smaller quantity of accumulated (++) species in this sample (Figure 2c), one could assume that these species are Fe-centred consistent with the smaller concentration of iron present. However, the concentration of Ni(Fe)OOH(++) states is at least twice the total concentration of Fe in the sample (see ESI and Table 1, entries 3 and 13), indicating a greater probability that these Ni(Fe)OOH(++) states are centred on Ni atoms. Furthermore, additional analysis of an NiOOH electrocatalyst kept free of any Fe impurities resulted in similar spectral features, thus effectively ruling-out any involvement of Fe centres in MOOH(++) species accumulation in samples with low concentrations of iron. This is in agreement with the detection of Ni⁴⁺ (i.e. NiOOH(++) in our nomenclature) in XAS and XANES experiments on this type of catalyst.^{27,28} ¹⁸O-labeling experiments by Hu and coworkers also indicate different active sites for Ni and NiFe oxides (Fe $\approx 25\%$),⁵² as do light scattering experiments on similar samples by Dau and coworkers, in-line with our conclusions herein.⁵³ However, the Fe concentration at which the favoured MOOH(++) species accumulated switches from Ni-centred to Fe-centred remains to be determined. It also cannot be ascertained from these data alone whether a regime exists in which both metals are responsible for charge accumulation, as reported by others.^{30,32,54} A degree of synergism between metal centres is probable, considering the improved charge accumulation found in the FeOOHNiOOH sample

over the FeOOH sample. More experiments are being carried out in our group in order to shed light on this question."

2. NiOOHFeOOH. I am not sure what this sample is. Are you sure they are phase separated? You show pre-catalyst separation. But how about after activation? The work of Bell et al. (JACS 2015) showed that at high Fe level, one would make NiFeOx plus FeOOH. Are you sure you are not making it? The paper emphasizes the "interface" between NiOOH and FeOOH. But one need to make sure this phase separation dominates.

We note that there is a distinctive difference between the method used by Bell et al. (JACS 2015) and our method. Bell et al. co-deposited FeOOH and NiOOH. Thus, at high Fe level their method resulted in the formation of NiFeOx, where Ni and Fe are mixed in the same phase, and pure FeOOH. On the other hand, our FeOOHNiOOH sample is prepared by sequential depositions, deposition of FeOOH followed by the deposition of NiOOH. Although NiOOH was deposited on top of FeOOH, due to the porous nature of FeOOH, significant mixing of Fe and Ni was achieved at the FeOOH/NiOOH interface. We do not think the distribution of Fe and Ni in our samples can be affected significantly by electrochemical activation. The preparation and nature of FeOOHNiOOH is provided in detail in the main text (page 2, lines 42-44) and ESI (Section A, 5th paragraph).

3. Fig. 1. Please note the loading in the caption. The geometric activity seems much lower than typical NiFeOx, but it might be due to a low loading. Please clarify.

Since the caption of Figure 1 is already too long and we need to report both NiOOH and FeOOH loadings for multiple samples, the loadings of NiOOH and FeOOH in all samples used in this study are summarized in Table S1. We note that the samples used in this study are deposited in a flat FTO substrate as thin films (~100 nm) with low loadings of FeOOH and NiOOH. Thus, the OER performances of our samples may be lower than that of NiFeOx deposited as thicker films or deposited on a high surface area substrate.

4. When you report experiments at 2 ma/cm², please also report the potential. It looks that the authors largely neglect potential-dependent activity in this paper. See also below. I feel this might be overlooked - rate can be potential dependent in addition to concentration of resting/intermediates.

The overpotential required for 2.5 mA/cm² of current is provided in Table 1, entry 5.

5. The authors make important assumptions in the estimation of concentrations for oxidized and doubly oxidized species. I think it is better to explain the assumption in the main text.

To better explain some of the assumptions made in our analyses, and their validity, the following sentence have been added to the revised manuscript and supporting information:

(page 9, lines 4-7)

“This agreement in kinetics between our combined optical/electrochemical data (Figure 4a and Table 1, entry 8) and optical data alone (ESI, Figure S15 and S18 and Table 1, entries 11 and 12) strongly supports the validity of our analyses, as these two kinetic analyses employ very different experimental and data analysis procedures (see ESI for more discussion).”

(ESI, page 19)

“We note that the kinetic analysis given in Figure 4 in the main article relies upon accurate calculation of the extinction coefficient for the MOOH[++] species such that their concentration may be determined from optical data. On the other hand, the analysis presented here in Figures S15 to S18 does not require such calculations, and the tau values in these cases (presented in Table 1, entry 8) are obtained only by fitting the raw kinetic traces. As such, the agreement in the tau values obtained from these independent measurements increases confidence in our analyses.”

6. The kinetic analysis is an important contribution to the field. However, I feel the current analysis need to clarify a few points. (a) As the authors are aware of, UV-Vis spectroscopy can only catch species with distinguished optical absorption. Differential spectroscopy only detects obvious differences. Is it possible that the real intermediates cannot be seen? (b) You assume one electron for each species you see. What is the proof? In literature people tend to assign 1.7 for first oxidation in Ni. In any case, please discuss these issues. (c) More importantly, according to your analysis, you are not seeing the resting states, right? what is the percentage of doubly oxidized species in your film? This is important for later mechanistic analysis. (d) Fig. 4a neglects potential influence. Justified? An electrochemical reaction will depend somehow to potential in terms of rate, unless a very special condition applies. Please clarify and discuss. You study the decay off potential. Is this rate exactly match the current density? In other words, are you suggesting that once the doubly oxidized species is made, all reactions are pure chemical in nature? (e) How do you explain the difference of 4a and 4b in terms of order of TOF? If potential is not important, the trends should be the same, no?

With regards to point (a), certainly, given the temporal resolution limitations of our measurements, we are not able to observe all reactive intermediates, especially those with a high energy and short lifetime. The MOOH[++] species that we observe are those accumulated during catalysis, i.e. during the steady state of the reaction. We are also fortunate that almost all metal oxides give detectable absorption features between 400 and 1400 nm. To be more explicit, we have added the following text to the revised manuscript (page 13, line 2-3):

“the MOOH(++) states generated can be considered in equilibrium with much shorter lived reactive intermediates (which are not directly observed in this study)”

The [+] and [++] notation is not intended to denote absolute oxidation states. The first oxidation step, which we assign to the oxidation Ni(OH)₂ to NiOOH, can largely be considered pre-catalytic, and the number of electrons involved (though nominally 1), may be greater than 1. Regarding the second oxidation step to form MOOH[++] , we also cannot determine the absolute oxidation states, and thus the number of electrons involved. However, our rate-law calculations reveal that there is a 4th order dependence on the concentration of these accumulated species for current

production and, as water oxidation involves the exchange of 4 electrons, one can propose that each species alone has an oxidative power equivalent to 1 electron. Please see also our response to Reviewer 1's Comment 6 and the resultant changes made to the manuscript.

With regards to point (c), we do not believe that the [++] species observed are resting states, but rather the species accumulated under the rate-limiting step of water oxidation catalysis. Alternatively, the [+] species could be considered the resting state of the film. As per the reviewers request, we have added the number of [++] species as a percentage of the total number of corresponding metal atoms in the film, in Table S2 in the revised Supporting Information (please see our response to Comment 6 from Reviewer 1).

In order to obtain the different concentrations of accumulated MOOH[++] species (given as the x-axis in Figure 4a), different stepped-potentials were applied. As such, this data very much incorporates potential influence. Regarding how the tau values are calculated, we do indeed measure the decay in potential which we have shown in several of our works on photoanode systems to be analogous to the current decay. As the doubly oxidised species is that accumulated prior to the rate-determining step, it is possible that intermediate steps after this are electronic as well as chemical in nature.

The solid circle data in Figure 4a and 4b is identical, with 4a plotted in terms of tau with concentration, and 4b in terms of TOF with overpotential. The trends are indeed the same, with one plotted as the inverse of the other.

7. Please add TOF at 300 mV in Table 1.

Unfortunately we are not able to add the TOF values at 300 mV of overpotential because the FeOOH electrocatalyst does not have a measurable turnover at this overpotential (see Figure 4b). We have instead added the TOF at ~340 mV to Table 1, entry 10.

8. In my opinion the section on Mechanistic discussion lacks solid support and limits the impact of the experimental work. As the authors acknowledge, they see only an intermediate before the RDS. They cannot probe the chemical nature. Moreover, they can only follow the decay of this species, while not being able to track proton transfer etc. Their discussion also does not reconcile with literature data in electrokinetics. They say "this disparity is can be attributed to the non-ideal nature of charge accumulation in these materials, with the different Tafel slopes most likely originating from differences in the dependence of charge density upon applied potential (Figure 2d), rather than being indicative of different reaction mechanisms." What does it mean non-ideal nature of charge accumulation? I have no idea. Keeping in mind Tafel studies include terms on potential dependence, while the data shown by the authors don't. I agree simple Tafel analysis is not easy to give a mechanism, but the contradiction with Tafel data is not easily dismissed.

The reviewer is correct in stating that we only observe the species accumulated before that RDS. Therefore, any rate that we observe with respect to the decay of these species can only give a lower limit to the rate of water oxidation catalysis, i.e. it does not express the turnover of any reactive intermediates, informing us only of the

slowest step (and therefore likely the best to target for enhancing catalysis). To make this clearer, we have added the following to the revised version of the manuscript (page 8, lines 26-29):

"We note that our studies determine the kinetics of the MOOH(++) species accumulated during steady state water oxidation catalysis, but cannot resolve the kinetics of any shorter lived reactive intermediates in the catalytic cycle. "

As the reviewer quotes from the manuscript, we do see a disparity between our results and those reported in the literature for Tafel type analyses. As mentioned in response to comment 6, our data does not neglect potential dependence on kinetics as each data point represents a different applied potential step. The difference, then, between our analyses and those of Tafel plots is in the inherent treatment of the material. Tafel analyses are more ideal for metals or similar material with an evenly distributed density of states that fill evenly with applied potential. However, the non-ideality that we referred to in the manuscript is the non-linear density of states accessible with applied potential that we see in these semiconductor materials. We have modified this in the revised manuscript to read as follows (page 11, lines 11-15):

"This disparity can be attributed to charge accumulation in these materials corresponding to multiple, localized redox oxidations at sites of direct electrolyte contact and therefore not behaving as ideal metals. As such, the different Tafel slopes most likely originate from differences in the dependence of charge density upon applied potential (Figure 2d), rather than being indicative of different reaction mechanisms."

9. The mechanism plots in Fig 5 are actually wrong. You should have only 4 e for OER, not 0 e as drawn. The addition of 4 e in RDS is strange. In any case, the data shown in the paper don't tell much about the mechanism. I recommend to mostly remove this section except keeping the discussion of "the nature of states" paragraph.

We agree with the reviewer that our labelling of the schematic in Figure 5 was perhaps confusing and have rectified this in the revised manuscript. We hope that by answering the above queries of the reviewer regarding our mechanism analysis, it now appears insightful to maintain this section in the manuscript. While this study does not go deep into mechanistic evaluation, we believe this schematic neatly provides a pictorial idea of how the 4th order rate law that we observe may be realised.

Reviewer #3 (Remarks to the Author):

The authors report a kinetic study of the water oxidation catalysis by nickel-iron oxyhydroxides. The main conclusions are that four equivalents of oxidized species are required for turnover to occur, that the kinetics have a fourth-order dependence on oxidized species, and that iron in itself is not responsible for the observed oxidized species. This is a well-performed study which complements the considerable data available on the iron-nickel system and is particularly useful in that it compares FeOOH and iron-rich mixed oxyhydroxide materials. Ultimately, however, the conclusions drawn are tenuously supported by the data. The paper would be more useful if it presented the data with all likely interpretations, rather than making a case that isn't there. There are a few peculiar findings of the study which require a more detailed treatment prior to publication, and I think that the conclusions relating to the role of iron centers is overstated and unsupported generally.

We thank the reviewer for their comments and hope that they find the revised manuscript satisfactory in supporting the conclusions made therein.

To begin with, the authors assign the absorption features in Figure 2a,b to d-d interband transitions and cites reference 27. If you look at reference 27, that assignment is just a throwaway - there is no justification for the assignment. Metal oxides and hydroxides like this tend to be insulators because, even though the d-orbitals aren't filled, they don't lead to conduction because the orbitals tend to be strongly localized at the metal centers; not delocalized into bands. So, the absorption features shown in Figure 2 are unlikely to be d-d absorptions, especially because of their intensities. It seems charge transfer is more likely. Because of this, it is difficult to draw the conclusion that any of the features are centered on a particular metal, which the authors use to strongly implicate nickel. I don't see this as a compelling argument.

We appreciate the concern regarding the nature of the transitions monitored by spectroelectrochemistry. Admittedly, it is not trivial to assign the transitions that give rise to the features observed without complementary analysis/energy computations, and we are more concerned with these features as markers for the different species found in each sample. That being said, d-d transitions are still probable as the absorption peaks in the non-normalised data are much less than $0.1 \Delta O.D$ (Figure S7). Even if the transitions were ligand to metal charge transfers, the difference in absorption between the Ni(Fe)OOH spectra and the spectra of FeOOH and FeOOHNiOOH can still be taken as a marker of the change in active metal centre, as the structure is otherwise comparable between the samples.

The Ni(Fe)OOH spectrum in Figure 2b is pretty interesting. Durrant in an earlier paper (Advances in Photoelectrochemical Water Splitting: Theory, Experiment and Systems Analysis, Chapter 5, available online) attributed broad 600 nm absorption features to Fe(IV)=O chromophores. This is in line with other studies which detect high-valent iron, but the authors do not consider this. Why?

Indeed, a broad absorption at a similar wavelength to that shown here in the Ni(Fe)OOH sample has been assigned to high-valent iron species in hematite. However, the main reason why we did not consider this species in our current manuscript is because it seems highly improbable that the Ni(Fe)OOH catalyst with only 5% incorporated Fe gives a high-valent iron oxo that is not observed in either FeOOH or FeOOHNiOOH. We now have further evidence to support the assignment

of the 650 nm peak in the Ni(Fe)OOH sample, as we have also prepared a NiOOH sample free of Fe impurities and measured the spectroelectrochemistry in Fe-free electrolyte. This is given in the revised manuscript in Figure S8. This NiOOH sample produces the same absorbance feature with no iron present, thereby ruling out any possibility for Fe(IV)=O chromophores.

The bar graph for Ni(Fe)OOH in Figure 2c would also be consistent with charge removal from Fe in the second (+/++) step, given its reduced amplitude relative to the charge removal at (0/+). This trend is not seen for the other materials, and is an important distinction.

This is a concept we had considered and ruled out once we deemed it unlikely that a small number of Fe-centres, distributed towards the edges of the material, were able to accumulate as much as half the number of oxidised species of the FeOOH sample, which has many more Fe-centres. With our new data considered, we are more certain that the MOOH[++] species accumulating under catalytic conditions in Ni(Fe)OOH are Ni-centred. However, we agree with the reviewer that this is worthy of mention and as such, have added to following to the revised manuscript (page 13, lines 18-25):

“Given the smaller quantity of accumulated (++) species in this sample (Figure 2c), one could assume that these species are Fe-centred consistent with the smaller concentration of iron present. However, the concentration of Ni(Fe)OOH(++) states is at least twice the total concentration of Fe in the sample (see ESI and Table 1, entries 3 and 13), indicating a greater probability that these Ni(Fe)OOH(++) states are centred on Ni atoms. Furthermore, additional analysis of an NiOOH electrocatalyst kept free of any Fe impurities resulted in similar spectral features, thus effectively ruling-out any involvement of Fe centres in MOOH(++) species accumulation in samples with low concentrations of iron.”

On p.12, the authors claim that the spectrum of Ni(Fe)OOH(++) doesn't change as a function of applied potential, and says this is an unlikely observation if the accumulation of 4 oxidative equivalents occurred on single isolated Fe centers distributed near the edges of the material. First, this is the appropriate place to cite and discuss Hunter et al. (Joule), and second, the authors need to explain why it is unlikely; I don't see this.

The sentence in question has now been removed in the revised manuscript due to increased certainty in the data. However, the rationale for the original statement is because if only the isolated Fe-centres were accumulating charge, it could be expected that they may increase in oxidation state more than once with increasing applied potential, and thus the absorption spectrum could be expected to change as new Fe-species are formed. No such change with bias is observed.

The fourth order rate law is a little curious as well. First, fourth order reactions do not happen in elementary steps. Second, I suspect that the rapid decay of the Ni(Fe)OOH(++) absorbance when the applied potential is removed, which is attributed to water oxidation, does not follow fourth-order kinetics. A fourth order reaction is protracted over a very long time period (compared to a first order reaction of comparable time constant). If the water oxidation rate depended on the fourth power of

[Ni(Fe)OOH(++)], then it would be very clear from the decays in S12 and S14. The kinetic fits in these figures is not convincing. Moreover, a plot of $1/[\text{Ni(Fe)OOH(++)}]^3$ should be linear. It is likely that the [Ni(Fe)OOH(++)] decay kinetics are not consistent with a fourth order reaction and, contrary to what the authors say, are not consistent with Figure 4a. What they measure in Figure 4a, is a steady state concentration of [Ni(Fe)OOH(++)] as a function of applied potential; their interpretation is the simple rate law in equation 1. But equation 1 embodies many elementary steps, and is unlikely to be consistent with the data in Figures S12 and S14.

The reviewer is of course correct that 4th order decays are very extended in time. For this reason, our decay data, and the fit to these data, are plotted on a logarithmic timescale to account for this, and reasonable fits are obtained to a 1st and 4th order model. The signal to noise on these decays is limited so these kinetic decays alone are not sufficient to reliably determine reaction order. For this reason, our rate law analyses are primarily based on our quasi-steady state analyses of (++) concentration and photocurrent density, as illustrated in Figure 4. A comment comparing both analyses has been added to the revised manuscript (page 9, lines 4-7):

“This agreement in kinetics between our combined optical/electrochemical data (Figure 4a and Table 1, entry 8) and optical data alone (ESI, Figure S15 and S18 and Table 1, entries 11 and 12) strongly supports the validity of our analyses, as these two kinetic analyses employ very different experimental and data analysis procedures (see ESI for more discussion).”

The reviewer is also correct that Equation 1 expresses only the overall rate law of the reaction, and does not include the underlying elementary steps in the catalytic mechanism. We have added additional comments to the manuscript emphasizing this (page 8, lines 26-28):

“We note that our studies determine the kinetics of the MOOH(++) species accumulated during steady state water oxidation catalysis, but cannot resolve the kinetics of any reactive intermediates in the catalytic cycle. .”

We note this same rate law equation is used to fit both the steady state data in Figure 4 and the decay data in Figure S18, further supporting the validity of our analyses (see comment above).

Based on the fact that the kinetics are not well resolved, some of the claims in the paper are dubious. On page 4, the authors note “[species] accumulated in Ni(Fe)OOH are distinct, with greater Ni character.” While it is clear that the features are distinct, what is the justification for claiming nickel character? This material is fundamentally different than the other materials synthesized, so why would sequential iron oxidation in this material look similar across all three? If these bands are indeed charge transfer bands, which is more likely than d-d transitions, wouldn't the local coordination environment of iron and nickel be the most pertinent difference? Similarly, the claim on page 8 line 13 that the states “probably [have] significant Ni character” is not supported by the discussion above in this light. This argument persists through the paper, concluding with page 12 line 18: “This would indicate that these... states are centered on Ni atoms, instead of on Fe.” A more robust analysis is required for these claims, since they do not seem to be uniquely supported by the data.

As mentioned in response to an earlier comment from the reviewer, we now have additional data to support our assignment to Ni-centred oxidised species, rather than isolated Fe-centres. This data, given in Figure S8, shows that the same distinct spectral feature is observed for an NiOOH sample synthesised and analysed in a purified electrolyte, free of Fe. With this new data, the several revisions have been made to the manuscript as detailed in the response to Reviewer 2, Comment 1.

An alternative conclusion from the data presented on page 9, lines 3-15 is that the Ni(Fe)OOH has fewer iron sites, resulting in less formation of the double oxidized species.

We are confident that this is not the case, as mentioned in the response above.

I am not convinced that the difference in Tafel slopes indicates the “non-ideal nature of charge accumulation” rather than a change in mechanism. Surely differences in charge accumulation will play a role in the Tafel slope, but in general vastly different Tafel slopes are not entirely attributable to this, rather to different underlying mechanisms.

The reviewer is correct that changes in Tafel slope are generally attributed to changes in the underlying mechanism. However, we mean to indicate this the Tafel plot that this method of kinetic analysis is ill-suited to these materials which present a non-flat/non-uniform density of states across varied potential. As with molecular systems, this non-ideality is problematic for traditional Tafel analyses. To strengthen this point, we have added the following to the revised manuscript (page 11, lines 12-15):

“This disparity can be attributed to charge accumulation in these materials corresponding to multiple, localized redox oxidations at sites of direct electrolyte contact and therefore not behaving as ideal metals. As such, the different Tafel slopes most likely originate from differences in the dependence of charge density upon applied potential (Figure 2d), rather than being indicative of different reaction mechanisms.”

The initial “charging” current needed to bring the catalyst to its resting state (generally agreed to be Ni(III)/Fe(III)) is well documented, so the discussion on page 7 (lines 1-8) should be tied in to the existing literature, rather than presented as a new discovery.

We have now rectified this with some rewording and additional references have been added. The revised manuscript now reads as follows (page 8, lines 6-11):

“Overall, these data confirm that the (0/+) process observed in our spectroelectrochemical data is an oxidation process which, though important for catalytic function, is not directly involved in water oxidation catalysis. This is in accordance with other documentation in the literature for such mixed metal oxides reporting an ‘activation’ or pre-catalytic step to bring the electrode to the catalytic resting state.^{27,45} Consequently, we focus hereafter only on the analysis of the (+/++) oxidation observed under conditions of water oxidation catalysis.”

There is a minor typographical error on page 6, line 5. The superscript on cm should be changed from 2 to -2.

This has been corrected in the revised manuscript.

REVIEWERS' COMMENTS:

Reviewer #1 (Remarks to the Author):

Considering the author's response to the comments of the review process, I feel that the quality of the manuscript has been greatly improved. Raised issues have been satisfactorily addressed according to my point of view. The manuscript in such provides a unique study that unifies several controversial discussions regarding the water oxidation kinetics in Ni-Fe oxyhydroxides earlier touched. There might of course still be doubts regarding the assignments of the spectral UV-vis features since no complementary technique has been applied, however, considering that the authors can show that the Fe-only catalyst has a redox peak hidden in the OER catalytic trace by studies in non-aqueous electrolytes, and taking into account that several other experimental and theoretical studies have confirmed redox-activity on the Fe-site, there is no need to further prove that this process is existing according to my opinion. I consider this manuscript a very interesting contribution to the scientific community, and might pave ground for future directions in time-resolved studies. I consider this manuscript suitable for publication in Nature Communications.

Reviewer #2 (Remarks to the Author):

The authors have done an excellent work in addressing the reviewers' concern. I am mostly satisfied with the response and changes. The paper is in principle ready for publication. However, I still have difficulty in agreeing with the way they present the mechanism. It seems all referees question how they use the 4th order pseudo-rate law to support Fig. 5, but the authors insisted. My main concern is really catalytic relevance. What the authors have observed is that the ++ species they identified, decay in an apparent 4th order. They acknowledge they cannot label them as either resting state, or the exact species before RDS. But then they propose four of such species will accumulate and give OER - implying although not explicitly, and these accumulated species make O-O bonds. I have several problems with this. First of all, there are many other possibilities in which these accumulated species are transformed into unidentified species which make O-O bond, so the Figure 5 is really misleading. Secondly, 4 e transfer is not necessarily parallel but can also be sequential - which is not considered in Figure 5. Thirdly, taking Ni(Fe)OOH, the observed accumulated species is Ni in nature, but in a very low concn. It is possible the Ni then pass the charge to Fe, in a way your techniques cannot detect but also cannot exclude, to make O₂. Again your Figure 5 is mis-leading. There are other issues for example, you are suggesting 4 sites work together to make O₂ in Figure 5. It is possible but probably unlikely. All in all, my point is that the data and findings are nice, but they seem to be compromised by the mechanism in Figure 5. I would suggest presenting various possibilities as I mentioned, maybe together with your possibility, so that your study facilitates mechanistic study, instead of concluding it.

Reviewer #3 (Remarks to the Author):

The authors have made a Herculean effort to address the (excellent) comments of the referees. While I still have doubts about the fourth-order rate law and the strength of the argument that the (+) and (++) species are unequivocally nickel, I think that the authors have justifiably added statements supporting their interpretation while not being heavy-handed.

I support the publication of this manuscript in its present form.

REVIEWERS' COMMENTS:

Reviewer #1 (Remarks to the Author):

Considering the author's response to the comments of the review process, I feel that the quality of the manuscript has been greatly improved. Raised issues have been satisfactorily addressed according to my point of view. The manuscript in such provides a unique study that unifies several controversial discussions regarding the water oxidation kinetics in Ni-Fe oxyhydroxides earlier touched. There might of course still be doubts regarding the assignments of the spectral UV-vis features since no complementary technique has been applied, however, considering that the authors can show that the Fe-only catalyst has a redox peak hidden in the OER catalytic trace by studies in non-aqueous electrolytes, and taking into account that several other experimental and theoretical studies have confirmed redox-activity on the Fe-site, there is no need to further prove that this process is existing according to my opinion. I consider this manuscript a very interesting contribution to the scientific community, and might pave ground for future directions in time-resolved studies. I consider this manuscript suitable for publication in Nature Communications.

We thank the reviewer for their kind comments on our revised manuscript and we also feel that the manuscript has improved through the reviewing process. We agree, as mentioned in our previous response, that a complementary probing technique could further add to our study and hope that others will follow suit with further investigations of their own.

Reviewer #2 (Remarks to the Author):

The authors have done an excellent work in addressing the reviewers' concern. I am mostly satisfied with the response and changes. The paper is in principle ready for publication.

We thank the reviewer for their acknowledgement of our efforts to address all comments made, and for reaching the conclusion that the manuscript is ready for publication.

However, I still have difficulty in agreeing with the way they present the mechanism. It seems all referees question how they use the 4th order pseudo-rate law to support Fig. 5, but the authors insisted. My main concern is really catalytic relevance. What the authors have observed is that the ++ species they identified, decay in an apparent 4th order. They acknowledge they cannot label them as either resting state, or the exact species before RDS. But then they propose four of such species will accumulate and give OER - implying although not explicitly, and these accumulated species make O-O bonds. I have several problems with this.

The reviewer is correct that there are several limitations to our measurements which do not allow us to probe each step in the catalytic cycle – we are only able to observe the species that accumulate under steady-state catalysis. As such, these species should be those before the rate determining step, and thus we were also surprised to find that these species give a 4th order rate law. We have tried to illustrate how this could come about in the mechanistic scheme in Figure 5. As rightly acknowledged by the reviewer, this is not a complete mechanism as we are unable to determine the active sites for bond making and breaking, but 4 of the observed species must come together in the RDS for us to observe the 4th order dependence.

First of all, there are many other possibilities in which these accumulated species are transformed into unidentified species which make O-O bond, so the Figure 5 is really misleading.

To try to clarify that we are not stating that the species we observe is involved directly in O-O bond making (though this is a possibility), we have added to our mechanistic Figure 5 to show that we cannot conclusive state the exact role of the accumulating species that we observe.

Secondly, 4 e transfer is not necessary parallel but can also be sequential - which is not considered in Figure 5.

As our rate laws for each electrocatalyst are 4th order, it is not instructive to show sequential loss of this accumulated population. Furthermore, we have no proof of this.

Thirdly, taking Ni(Fe)OOH, the observed accumulated species is Ni in nature, but in a very low concentration. It is possible the Ni then pass the charge to Fe, in a way your techniques cannot detect but also cannot exclude, to make O₂. Again your Figure 5 is misleading. There are other issues for example, you are suggesting 4 sites work together to make O₂ in Figure 5. It is possible but probably unlikely.

It is indeed possible that Fe is still involved in the O-O bond forming step of water oxidation catalysis in Ni(Fe)OOH, even though we find that the accumulated species observed are Ni-centred. This is why the individual steps are water oxidation catalysis are not shown, only the accumulation and how this fits into catalysis. Our mechanistic insight highlights that the accumulation of 4 holes for water oxidation is not rate

limiting, but that the process after this accumulation is slowing catalysis. Whether that is the final transfer of 4 accumulated charges to one active metal or the final O-O bond making step, we cannot determine.

All in all, my point is that the data and findings are nice, but they seem to be compromised by the mechanism in Figure 5. I would suggest presenting various possibilities as I mentioned, maybe together with your possibility, so that your study facilitates mechanistic study, instead of concluding it.

We hope that the reviewer will now be satisfied with our addition of uncertainty to the revised Figure 5 mechanism.

Reviewer #3 (Remarks to the Author):

The authors have made a Herculean effort to address the (excellent) comments of the referees. While I still have doubts about the fourth-order rate law and the strength of the argument that the (+) and (++) species are unequivocally nickel, I think that the authors have justifiably added statements supporting their interpretation while not being heavy-handed.

I support the publication of this manuscript in its present form.

We are grateful for the kind words of the reviewer and their insights on our manuscript.